# ON THE DISCRIMINATION-GENERALIZATION TRADE-OFF IN GANS

**Pengchuan Zhang**
Microsoft Research, Redmond
penzhan@microsoft.com

**Qiang Liu**
Computer Science, Dartmouth College
qiang.liu@dartmouth.edu

**Dengyong Zhou**
Google
dennyzhou@google.com

**Tao Xu**
Computer Science, Lehigh University
tax313@lehigh.edu

**Xiaodong He**
Microsoft Research, Redmond
xiaohe@microsoft.com

## ABSTRACT

Generative adversarial training can be generally understood as minimizing certain moment matching loss defined by a set of discriminator functions, typically neural networks. The discriminator set should be large enough to be able to uniquely identify the true distribution (*discriminative*), and also be small enough to go beyond memorizing samples (*generalizable*). In this paper, we show that a discriminator set is guaranteed to be discriminative whenever its linear span is dense in the set of bounded continuous functions. This is a very mild condition satisfied even by neural networks with a *single neuron*. Further, we develop generalization bounds between the learned distribution and true distribution under different evaluation metrics. When evaluated with neural distance, our bounds show that generalization is guaranteed as long as the discriminator set is small enough, regardless of the size of the generator or hypothesis set. When evaluated with KL divergence, our bound provides an explanation on the counter-intuitive behaviors of testing likelihood in GAN training. Our analysis sheds lights on understanding the practical performance of GANs.

## 1 INTRODUCTION

Generative adversarial networks (GANs) (Goodfellow et al., 2014) and their variants can be generally understood as minimizing certain moment matching loss defined by a set of discriminator functions. Mathematically, GANs minimize the integral probability metric (IPM) (Müller, 1997), that is,

$$\min_{\nu \in \mathcal{G}} \left\{ d_{\mathcal{F}}(\hat{\mu}_m, \nu) := \sup_{f \in \mathcal{F}} \left\{ \mathbb{E}_{x \sim \hat{\mu}_m}[f(x)] - \mathbb{E}_{x \sim \nu}[f(x)] \right\} \right\}, \tag{1}$$

where $\hat{\mu}_m$ is the empirical measure of the observed data, and $\mathcal{F}$ and $\mathcal{G}$ are the sets of discriminators and generators, respectively.

1. **Wasserstain GAN (W-GAN)** (Arjovsky et al., 2017). $\mathcal{F} = \text{Lip}_1(X) := \{f : ||f||_{\text{Lip}} \leq 1\}$, corresponding to the Wasserstain-1 distance.

2. **MMD-GAN** (Li et al., 2015; Dziugaite et al., 2015; Li et al., 2017a). $\mathcal{F}$ is taken as the unit ball in a Reproducing Kernel Hilbert Space (RKHS), corresponding to the Maximum Mean Discrepency (MMD).

3. **Energy-based GANs** (Zhao et al., 2016). $\mathcal{F}$ is taken as the set of continuous functions bounded between 0 and $M$ for some constant $M > 0$, corresponding to the total variation distance (Arjovsky et al., 2017).

4. **f-GAN** (Nowozin et al., 2016) minimizes the $f$-divergence, which can be viewed a form of regularized moment matching loss defined over all possible functions as shown by Liu et al. (2017). See also Appedix B.

Due to computational tractability, however, the practical GANs take $\mathcal{F}$ as a parametric function class, typically, $\mathcal{F}_{\mathrm{nn}} = \{f_\theta(x) \colon \theta \in \Theta\}$ where $f_\theta(x)$ is a neural network indexed by parameters $\theta$ that take values in $\Theta \subset \mathbb{R}^p$. Consequently, the related $d_{\mathcal{F}_{\mathrm{nn}}}(\mu, \nu)$ is called neural network distance, or neural distance (Arora et al., 2017). Although $d_{\mathcal{F}_{\mathrm{nn}}}(\mu, \nu)$ is meant to be a surrogate, its properties can be fundamentally different from the original objective functions. For example, in W-GAN, because $\mathcal{F}_{\mathrm{nn}}$ is a much smaller discriminator set than $\mathrm{Lip}_1(X)$, it is unclear from the current GAN literature whether $d_{\mathcal{F}_{\mathrm{nn}}}(\mu, \nu)$ is a discriminative metric in that $d_{\mathcal{F}_{\mathrm{nn}}}(\mu, \nu) = 0$ implies $\mu = \nu$. This discrimination is critical to ensure the consistency of the learning result. This motivated us to study the properties of $d_{\mathcal{F}_{\mathrm{nn}}}(\mu, \nu)$ with parametric function sets $\mathcal{F}_{\mathrm{nn}}$, instead of the original Wasserstein distance or $f$-divergence.

A broader question is in developing learning bounds and studying how they depend on the discriminator set $\mathcal{F}$ and the generator set $\mathcal{G}$, under different evaluation metrics of interest. Specifically, assuming $\nu_m$ is an (approximate) solution of (1), we are interested in obtaining bounds between $\nu_m$ and the underlying true distribution $\mu$, under a given evaluation metric $d_{\mathrm{eval}}(\mu, \nu_m)$. Existing analysis has been mostly focusing on the case when the evaluation metric coincides with the optimization metric, that is, $d_{\mathrm{eval}}(\mu, \nu) = d_{\mathcal{F}}(\mu, \nu)$, which, however, favors smaller discriminator sets that define *"easier"* evaluation metrics. It is of interest to develop bounds for evaluation metrics independent of $\mathcal{F}$, such as bounded Lipschitz distance that metrizes weak convergence, and KL divergence that connects to testing likelihood.

**Contribution.** We show that the role of discriminators $\mathcal{F}$ is best illustrated by the conditions under which $d_{\mathcal{F}}(\mu, \nu)$ metrizes weak convergence (or convergence in distribution), that is,

$$d_{\mathcal{F}}(\mu, \nu_m) \to 0 \quad \text{if and only if} \quad \nu_m \rightharpoonup \mu, \tag{2}$$

for any probability measures $\mu$ and $\nu_m$. The choice of $\mathcal{F}$ should strike a balance to achieve (2):

i) $\mathcal{F}$ should be large enough to make $d_{\mathcal{F}}(\mu, \nu)$ *discrimiantive* in that $d_{\mathcal{F}}(\mu, \nu_m) \to 0$ can imply that $\nu_m$ weakly converges to $\mu$. Further, with a given metric $d_{\mathrm{eval}}(\mu, \nu)$, the discriminator set $\mathcal{F}$ should be large enough so that a small $d_{\mathcal{F}}(\mu, \nu)$ implies a small $d_{\mathrm{eval}}(\mu, \nu)$ in certain sense. These are basic requirements in justifying $d_{\mathcal{F}}(\mu, \nu)$ as a valid learning objective function.

ii) $\mathcal{F}$ should also be relatively small so that $\nu_m \rightharpoonup \mu$ implies that $d_{\mathcal{F}}(\mu, \nu_m)$ approaches to zero. This is essential to guarantee that the training and testing loss are similar to each other and hence the algorithm is *generalizable*. Further, in order to obtain a low sample complexity, $\mathcal{F}$ should be sufficiently small so that $d_{\mathcal{F}}(\mu, \nu_m)$ decays with a fast rate, preferably $O(1/\sqrt{m})$.

The theme of this work is to characterize the conditions under which i) and ii) hold and develop bounds of $d_{\mathrm{eval}}(\mu, \nu_m)$ that characterize the role of discriminators $\mathcal{F}$ and generators $\mathcal{G}$. Our contributions are summarized as follows.

1. We show that a discriminator set $\mathcal{F}$ is discriminative once the linear span of $\mathcal{F}$ is dense in the bounded continuous (or Lipschitz) function space. This is a mild condition that can satisfied, for example, even for neural networks consists of a *single neuron*. See Section 2.

2. We develop techniques using neural distance $d_{\mathcal{F}}(\mu, \nu)$ to provide upper bounds of different evaluation metrics $d_{\mathrm{eval}}(\mu, \nu)$ of interest, including bounded Lipschitz (BL) distance and KL divergence, which provides a key step for developing learning bounds of GANs under these metrics. See Section 2.1.

3. We characterize the generalizability of GANs using the Rademacher complexity of discriminator set $\mathcal{F}$ and put together bounds between the true distributions $\mu$ and GAN estimators $\nu_m$ under different evaluation metrics in Section 3. Under the neural distance, our bounds (Corollary 3.2-3.3) show that generalization is guaranteed as long as the discriminator set is small enough, *regardless of the size of the generator or hypothesis set $\mathcal{G}$*. This seemingly-surprising result is reasonable because in this case the evaluation metric $d_{\mathcal{F}}(\mu, \nu_m)$ depends on the discriminator set.

4. When the KL divergence is used as the evaluation metric, our bound (Corollary 3.5) suggests that the generator and discriminator sets have to be compatible in that the log density ratios of the generators and the true distributions should exist and be included inside the linear span of the discriminator set. The strong condition that log-density ratio should exist partially explains the counter-intuitive behavior of testing likelihood in flow GANs (e.g., Danihelka et al., 2017; Grover et al., 2017).

5. We extend our analysis to study neural $f$-divergences that are the learning objective of $f$-GANs, and establish similar results on the discrimination and generalization properties of neural $f$-divergences; see Appendix B. Different from neural distance, a neural $f$-divergence is discriminative if linear span of its discriminators *without the output activation function* is dense in the bounded continuous function space.

## 1.1 NOTATIONS

We use $X$ to denote a subset of $\mathbb{R}^d$. For each continuous function $f\colon X \to \mathbb{R}$, we define the maximum norm as $\|f\|_\infty = \sup_{x \in X} |f(x)|$, and the Lipschitz norm $\|f\|_{\mathrm{Lip}} = \sup\{|f(x) - f(y)|/\|x - y\|\colon x, y \in X, \ x \neq y\}$, and the bounded Lipschitz (BL) norm $\|f\|_{\mathrm{BL}} = \max\{\|f\|_{\mathrm{Lip}}, \|f\|_\infty\}$. The set of continuous functions on $X$ is denoted by $C(X)$, and the Banach space of bounded continuous function is $C_b(X) = \{f \in C(X)\colon \|f\|_\infty < \infty\}$.

The set of Borel probability measures on $X$ is denoted by $\mathcal{P}_\mathcal{B}(X)$. In this paper, we assume that all measures involved belong to $\mathcal{P}_\mathcal{B}(X)$, which is sufficient in all practical applications. We denote by $\mathbb{E}_\mu[f]$ the integral of $f$ with respect to probability measure $\mu$. The weak convergence, or convergence in distribution, is denoted by $\nu_n \rightharpoonup \nu$. Given a base measures $\tau$ (e.g., Lebesgue measure), the density of $\mu \in \mathcal{P}_\mathcal{B}(X)$, if it exists, is denoted by $\rho_\mu = \frac{\mathrm{d}\mu}{\mathrm{d}\tau}$. We do *not* assume density exists in our main theoretical results, except the cases when discussing KL divergence.

## 2 DISCRIMINATIVE PROPERTIES OF NEURAL DISTANCES FOR GAN

As listed in the introduction, many variants of GAN can be viewed as minimizing the integral probability metric (1). Without loss of generality, we assume that the discriminator set $\mathcal{F}$ is even, i.e., $f \in \mathcal{F}$ implies $-f \in \mathcal{F}$. Intuitively speaking, minimizing (1) towards zero corresponds to matching the moments $\mathbb{E}_\mu[f] = \mathbb{E}_\nu[f]$ for all discriminators $f \in \mathcal{F}$. In their original formulation, all those discriminator sets are non-parametric, infinite dimensional, and large enough to guarantee that $d_\mathcal{F}(\mu, \nu) = 0$ implies $\mu = \nu$.

In practice, however, the discriminator set is typically restricted to parametric function classes of form $\mathcal{F}_{\mathrm{nn}} = \{f_\theta\colon \theta \in \Theta\}$. When $f_\theta$ is a neural network, we call $d_{\mathcal{F}_{\mathrm{nn}}}(\mu, \nu)$ a neural distance following Arora et al. (2017). Neural distances are the actual object function that W-GAN optimizes in practice because they can be practically optimized and can leverage the representation power of neural networks. Therefore, it is of great importance to directly study neural distances, instead of Wasserstein metric, in order to understand practical performance of GANs.

Because the parameter function set $\mathcal{F}_{\mathrm{nn}}$ is much smaller than the non-parametric sets like $\mathrm{Lip}_1(X)$, a key question is whether $\mathcal{F}_{\mathrm{nn}}$ is large enough so that moment matching on $\mathcal{F}_{\mathrm{nn}}$ (i.e., $d_{\mathcal{F}_{\mathrm{nn}}}(\mu, \nu) = 0$) implies $\mu = \nu$. It turns out the answer is affirmative once $\mathcal{F}_{\mathrm{nn}}$ is large enough so that its *linear span* (instead of $\mathcal{F}_{\mathrm{nn}}$ itself) forms a universal approximator. This is a rather weak condition, which is satisfied even by very small sets such as neural networks with a *single neuron*.

We make this concrete in the following.
*Definition* 2.1. Let $(X, d_X)$ be a metric space and $\mathcal{F}$ be a set of functions on $X$. We say that $d_\mathcal{F}(\mu, \nu)$ (and $\mathcal{F}$) is discriminative if

$$d_\mathcal{F}(\mu, \nu) = 0 \iff \mu = \nu,$$

for any two Borel probability measures $\mu, \nu \in \mathcal{P}_\mathcal{B}(X)$. In other words, $\mathcal{F}$ is discriminative if the moment matching on $\mathcal{F}$, i.e., $\mathbb{E}_\mu[f] = \mathbb{E}_\nu[f]$ for any $f \in \mathcal{F}$, implies $\mu = \nu$.

The key observation is that $\mathbb{E}_\mu[f] = \mathbb{E}_\nu[f]$ for any $f \in \mathcal{F}$ implies the same holds true for all $f$ in the linear span of $\mathcal{F}$. Therefore, it is sufficient to require the linear span of $\mathcal{F}$, instead of $\mathcal{F}$ itself, to be large enough to well approximate all the indicator test functions.

**Theorem 2.2.** *For a given function set $\mathcal{F} \subset C_b(X)$, define*

$$span\mathcal{F} := \{\alpha_0 + \sum_{i=1}^{n} \alpha_i f_i : \alpha_i \in \mathbb{R}, f_i \in \mathcal{F}, n \in \mathbb{N}\}. \tag{3}$$

*Then $d_{\mathcal{F}}(\mu, \nu)$ is discriminative if $span\mathcal{F}$ is dense in the space of bounded continuous functions $C_b(X)$ under the uniform norm $\|\cdot\|_\infty$, that is, for any $f \in C_b(X)$ and $\epsilon > 0$, there exists an $f_\epsilon \in span\mathcal{F}$ such that $\|f - f_\epsilon\|_\infty \leq \epsilon$. An equivalent way to put is that $C_b(X)$ is included in the closure of $span\mathcal{F}$, that is,*

$$cl(span\mathcal{F}) \supseteq C_b(X). \tag{4}$$

*Further, (4) is a necessary condition for $d_{\mathcal{F}}(\mu, \nu)$ to be discriminative if $X$ is a compact space.*

**Remark 2.1.** *The basic idea of characterizing probability measures using functions in $C_b(X)$ is closely related to the concept of weak convergence. Recall that a sequence $\nu_n$ weakly converges to $\mu$, i.e., $\nu_n \rightharpoonup \mu$, if and only if $\mathbb{E}_{\nu_n}[f] \to \mathbb{E}_\mu[f]$ for all $f \in C_b(X)$. Proof of the sufficient part of Theorem 2.2 is standard and the same with proof of the uniqueness of weak convergence; see, e.g., Lemma 9.3.2 in Dudley (2002).*

**Remark 2.2.** *We obtain similar results for neural $f$-divergence $d_{\phi, \mathcal{F}}(\mu \| \nu)$ in Theorem B.1. The difficulty in analyzing neural $f$-divergence is that moment matching on the discriminator set is only a sufficient condition for minimizing neural $f$-divergence, i.e.,*

$$\{\nu : \mathbb{E}_\mu[f] = \mathbb{E}_\nu[f], \forall f \in \mathcal{F}\} \subseteq \left\{\nu : d_{\phi, \mathcal{F}}(\mu \| \nu) = \arg\min_\nu d_{\phi, \mathcal{F}}(\mu \| \nu)\right\}.$$

*Consequently, $cl(span\mathcal{F}) \supseteq C_b(X)$ is only necessary but not sufficient for a neural $f$-divergence to be discriminative. When the discriminators are neural networks, we show that moment matching on the function set that consists of discriminators without their output activation function, denoted as $\mathcal{F}_0$, is a necessary condition for minimizing neural $f$-divergence, i.e.,*

$$\left\{\nu : d_{\phi, \mathcal{F}}(\mu \| \nu) = \arg\min_\nu d_{\phi, \mathcal{F}}(\mu \| \nu)\right\} \subseteq \{\nu : \mathbb{E}_\mu[f_0] = \mathbb{E}_\nu[f_0], \forall f_0 \in \mathcal{F}_0\}.$$

*We refer to Theorem B.1 (ii) for a precise statement. Therefore, a neural $f$-divergence is discriminative if linear span of its discriminators without the output activation function is dense in the bounded continuous function space.*

**Remark 2.3.** *Because the set of bounded Lipschitz functions $\mathrm{BL}(X) = \{f \in C_b(X) : \|f\|_{\mathrm{Lip}} < \infty\}$ is dense in $C_b(X)$, the condition in (4) can be replaced by a weaker condition $cl(span\mathcal{F}) \supseteq \mathrm{BL}(X)$. One can define a norm $\|\cdot\|_{\mathrm{BL}}$ for functions in $\mathrm{BL}(X)$ by $\|f\|_{\mathrm{BL}} = \max\{\|f\|_{\mathrm{Lip}}, \|f\|_\infty\}$. This defines the bounded Lipschitz (BL) distance,*

$$d_{\mathrm{BL}}(\mu, \nu) = \max_{f \in \mathrm{BL}(X)} \{\mathbb{E}_\mu f - \mathbb{E}_\nu f : \|f\|_{\mathrm{BL}} \leq 1\}.$$

*The BL distance is known to metrize weak convergence in sense that $d_{\mathrm{BL}}(\mu, \nu_n) \to 0$ is equivalent to $\nu_n \rightharpoonup \mu$ for all Borel probability measures on $\mathbb{R}^d$; see section 8.3 in Bogachev (2007).*

**Neural distances are discriminative.** The key message of Theorem 2.2 is that it is sufficient to require $cl(span\mathcal{F}) \supseteq C_b(X)$ (Condition (4)), which is a much weaker condition than the perhaps more straightforward condition $cl(\mathcal{F}) \supseteq C_b(X)$. In fact, (4) is met by function sets that are much smaller than what we actually use in practice. For example, it is satisfied by the neural networks with only *a single* neuron, i.e.,

$$\mathcal{F}_{\mathrm{nn}} = \{\sigma(w^\top x + b) : w \in \mathbb{R}^d, b \in \mathbb{R}\}. \tag{5}$$

This is because its span $span\mathcal{F}_{\mathrm{nn}}$ includes neural networks with infinite numbers of neurons, which are well known to be universal approximators in $C_b(X)$ according to classical theories (e.g., Cybenko, 1989; Hornik et al., 1989; Hornik, 1991; Leshno et al., 1993; Barron, 1993). We recall the following classical result.

**Theorem 2.3** (Theorem 1 in Leshno et al. (1993)). *Let $\sigma : \mathbb{R} \to \mathbb{R}$ be a continuous activation function and $X \subset \mathbb{R}^d$ be any compact set. Let $\mathcal{F}_{\mathrm{nn}}$ be the set of neural networks with a single neuron as defined in (5), then $span\mathcal{F}_{\mathrm{nn}}$ is dense in $C(X)$ if and only if $\sigma$ is not a polynomial.*

The above result requires that the parameters $[w, b]$ take values in $\mathbb{R}^{d+1}$. In practice, however, we can only efficiently search in bounded parameter sets of $[w, b]$ using local search methods like gradient descent. We observe that it is sufficient to replace $\mathbb{R}^{d+1}$ with a bounded parameter set $\Theta$ for non-decreasing homogeneous activation functions such as $\sigma(u) = \max\{u, 0\}^\alpha$ with $\alpha \in \mathbb{N}$; note that $\alpha = 1$ is the widely used rectified linear unit (ReLU).

**Corollary 2.4.** *Let $X \subset \mathbb{R}^d$ be any compact set, and $\sigma(u) = \max\{u, 0\}^\alpha$ ($\alpha \in \mathbb{N}$), and $\mathcal{F}_{\mathrm{nn}} = \{\sigma(w^\top x + b) \colon [w, b] \in \Theta\}$. Then $span\mathcal{F}_{\mathrm{nn}}$ is dense in $C_b(X)$ if*

$$\{\lambda\theta : \lambda \geq 0, \theta \in \Theta\} = \mathbb{R}^{d+1}.$$

For the case when $\Theta = \{\theta \in \mathbb{R}^{d+1} : \|\theta\|_2 \leq 1\}$, Bach (2017) not only proves that $span\mathcal{F}_{\mathrm{nn}}$ is dense in $\mathrm{Lip}_1(X)$ (and thus dense in $C_b(X)$), but also gives the convergence rate.

Therefore, for ReLU activation functions, $\mathcal{F}_{\mathrm{nn}}$ with bounded parameter sets, like $\{\theta : \|\theta\| \leq 1\}$ or $\{\theta : \|\theta\| = 1\}$ for any norm on $\mathbb{R}^{d+1}$, is sufficient to discriminate any two Borel probability measures. Note that this is not true for some other activation functions such as tanh or sigmoid, because there is an approximation gap between $span\{\sigma(w^\top x + b) : [w, b] \in \Theta \subset \mathbb{R}^{d+1}\}$ and $C_b(X)$ when $\Theta \subset \mathbb{R}^{d+1}$ is bounded; see e.g., Barron (1993) (Theorem 3). From this perspective, homogeneous activation functions such as ReLU are preferred as discriminators.

One advantage of using bounded parameter set $\Theta$ is that it makes $\mathcal{F}_{\mathrm{nn}}$ have a bounded Lipschitz norm, and hence the corresponding neural distance is upper bounded by Wasserstein distance. In fact, W-GAN uses weight clipping to explicitly enforce $\|\theta\|_\infty \leq \delta$. However, we should point out that the Lipschitz constraint does not help in making $\mathcal{F}$ discriminative since the constraint decreases, instead of enlarges, the function set $\mathcal{F}$. Instead, the role of the Lipschitz constraint should be mostly in stabilizing the training (Arjovsky et al., 2017) and assuring a generalization bound as we discuss in Section 3. Another related way to justify the Lipschitz constraint is its relation to metrizing weak convergence, as we discuss in the sequel.

**Neural distance and weak convergence.** If $\mathcal{F}$ is discriminative, then $d_{\mathcal{F}}(\mu, \nu) = 0$ implies $\mu = \nu$. In practice, however, we often cannot achieve $d_{\mathcal{F}}(\mu, \nu) = 0$ strictly. Instead, we often have $d_{\mathcal{F}}(\mu, \nu_n) \to 0$ for a sequence of $\nu_n$ and want to establish the weak convergence $\nu_n \rightharpoonup \mu$.

**Theorem 2.5.** *Let $(X, d_X)$ be any metric space. If $span\mathcal{F}$ is dense in $C_b(X)$, we have $\lim_{n\to\infty} d_{\mathcal{F}}(\mu, \nu_n) = 0$ implies $\nu_n$ weakly converges to $\mu$.*

*Additionally, if $\mathcal{F}$ is contained in a bounded Lipchitz function space, i.e., there exists $0 < C < \infty$ such that $\|f\|_{\mathrm{BL}} \leq C$ for all $f \in \mathcal{F}$, then $\nu_n$ weakly converges to $\mu$ implies $\lim_{n\to\infty} d_{\mathcal{F}}(\mu, \nu_n) = 0$.*

Theorem 10 of Liu et al. (2017) states a similar result for generic adversarial divergences, but does not obtain the specific weak convergence result for neural distances due to lacking of Theorem 2.2. Another difference is that Theorem 10 of Liu et al. (2017) heavily relies on the compactness assumption of $X$, while our result does not need this assumption. We provide the proof for Theorem 2.5 in Appendix C.

When $X$ is compact, Wasserstein distance and the BL distance are equivalent and both metrize weak convergence. As we discussed earlier, the condition $cl(span\mathcal{F}) = C_b(X)$ and $\mathcal{F} \subseteq \mathrm{Lip}_K(X)$ are satisfied by neural networks $\mathcal{F}_{\mathrm{nn}}$ with ReLU activation function and bounded parameter set $\Theta$. Therefore, the related neural distance $d_{\mathcal{F}_{\mathrm{nn}}}$ is topologically equivalent to the Wasserstein and BL distance, because all of them metrize the weak convergence. This *does not imply*, however, that they are equivalent in the metric sense (or strongly equivalent) since the ratio $d_{\mathrm{BL}}(\mu, \nu)/d_{\mathcal{F}_{\mathrm{nn}}}(\mu, \nu)$ can be unbounded. In general, the neural distances are weaker than the BL distance because of smaller $\mathcal{F}$. In Section 2.1 (and particularly Corollary 2.8), we draw more discussions on the bounds between BL distance and neural distances.

## 2.1 DISCRIMINATIVE POWER OF NEURAL DISTANCES

Theorem 2.2 characterizes the condition under which a neural distance is discriminative, and shows that even neural networks with a single neuron are sufficient to be discriminative. This does not explain, however, why it is beneficial to use larger and deeper networks as we do in practice. What is missing here is to frame and understand *how discriminative or strong* a neural distance is. This is

because even if $d_{\mathcal{F}}(\mu, \nu)$ is discriminative, it can be *relatively weak* in that $d_{\mathcal{F}}(\mu, \nu)$ may be small when $\mu$ and $\nu$ are very different under standard metrics (e.g., BL distance). Obviously, a larger $\mathcal{F}$ yields a stronger neural distance, that is, if $\mathcal{F} \subset \mathcal{F}'$, then $d_{\mathcal{F}}(\mu, \nu) \leq d_{\mathcal{F}'}(\mu, \nu)$. For example, because it is reasonable to assume that neural networks are bounded Lipschitz when $X$ and $\Theta$ are bounded, we can control a neural distance with the BL distance:

$$d_{\mathcal{F}}(\mu, \nu) \leq C d_{\mathrm{BL}}(\mu, \nu),$$

where $C := \sup_{f \in \mathcal{F}} \{||f||_{\mathrm{BL}}\} < \infty$. A more difficult question is if we can establish inequalities in the other direction, that is, controlling $d_{\mathrm{BL}}(\mu, \nu)$, or in general a stronger $d_{\mathcal{F}'}(\mu, \nu)$, with a weaker $d_{\mathcal{F}}(\mu, \nu)$ in some way. In this section, we characterize conditions under which this is possible and develop bounds that allow us to use neural distances to control stronger distances such as BL distance, and even KL divergence. These bounds are used in Section 3 to translate generalization bounds in $d_{\mathcal{F}}(\mu, \nu)$ to that in BL distance and KL divergence.

The core of the discussion involves understanding how $d_{\mathcal{F}}(\mu, \nu)$ can be used to control the difference of the moment $|\mathbb{E}_\mu\, g - \mathbb{E}_\nu\, g|$ for $g$ *outside of* $\mathcal{F}$. We address this problem by two steps: first controlling functions in $span\mathcal{F}$, and then functions in $cl(span\mathcal{F})$ that is large enough to include $C_b(X)$ for neural networks.

**Controlling functions in $span\mathcal{F}$.** We start with understanding how $d_{\mathcal{F}}(\mu, \nu)$ can bound $|\mathbb{E}_\mu\, g - \mathbb{E}_\nu\, g|$ for $g \in span\mathcal{F}$. This can be characterized by introducing a notion of norm on $span\mathcal{F}$.

**Proposition 2.6.** *For each $g \in span\mathcal{F}$ that can be decomposed into $g = \sum_{i=1}^n w_i f_i + w_0$ as we define in (3), the $\mathcal{F}$-variation norm $||g||_{\mathcal{F},1}$ of $g$ is the infimum of $\sum_{i=1}^n |w_i|$ among all possible decompositions of $g$, that is,*

$$||g||_{\mathcal{F},1} = \inf \left\{ \sum_{i=1}^n |w_i| : g = \sum_{i=1}^n w_i f_i + w_0, \; \forall n \in \mathbb{N}, \; w_0, w_i \in \mathbb{R}, \; f_i \in \mathcal{F} \right\}.$$

*Then we have*

$$|\mathbb{E}_\mu\, g - \mathbb{E}_\nu\, g| \leq ||g||_{\mathcal{F},1}\, d_{\mathcal{F}}(\mu, \nu), \quad \forall g \in span\mathcal{F}.$$

Intuitively speaking, $||g||_{\mathcal{F},1}$ denotes the "minimum number" of functions in $\mathcal{F}$ needed to represent $g$. As $\mathcal{F}$ becomes larger, $||g||_{\mathcal{F},1}$ decreases and $d_{\mathcal{F}}(\mu, \nu)$ can better control $|\mathbb{E}_\mu\, g - \mathbb{E}_\nu\, g|$. Precisely, if $\mathcal{F} \subseteq \mathcal{F}'$ then $||g||_{\mathcal{F}',1} \leq ||g||_{\mathcal{F},1}$. Therefore, although adding more neurons in $\mathcal{F}$ may not necessarily enlarge $span\mathcal{F}$, it decreases $||g||_{\mathcal{F},1}$ and yields a stronger neural distance.

**Controlling functions in $cl(span\mathcal{F})$.** A more critical question is how the neural distance $d_{\mathcal{F}}(\mu, \nu)$ can also control the discrepancy $\mathbb{E}_\mu\, g - \mathbb{E}_\nu\, g$ for functions outside of $span\mathcal{F}$ but inside $cl(span\mathcal{F})$. The bound in this case is characterized by a notion of error decay function defined as follows.

**Proposition 2.7.** *Given a function $g$, we say that $g$ is approximated by $\mathcal{F}$ with error decay function $\epsilon(r)$ if for any $r \geq 0$, there exists an $f_r \in span\mathcal{F}$ with $||f_r||_{\mathcal{F},1} \leq r$ such that $||f - f_r||_\infty \leq \epsilon(r)$. Therefore, $g \in cl(span\mathcal{F})$ if and only if $\inf_{r \geq 0} \epsilon(r) = 0$. We have*

$$|\mathbb{E}_\mu\, g - \mathbb{E}_\nu\, g| \leq \inf_{r \geq 0} \{2\epsilon(r) + r\, d_{\mathcal{F}}(\mu, \nu)\}.$$

*In particular, if $\epsilon(r) = O(r^{-\kappa})$ for some $\kappa > 0$, then $|\mathbb{E}_\mu\, g - \mathbb{E}_\nu\, g| = O(d_{\mathcal{F}}(\mu, \nu)^{\frac{\kappa}{\kappa+1}})$.*

It requires further efforts to derive the error decay function for specific $\mathcal{F}$ and $g$. For example, Proposition 6 of Bach (2017) allows us to derive the decay rate of approximating bounded Lipschitz functions with rectified neurons, yielding a bound between BL distance and neural distance.

**Corollary 2.8.** *Let $X$ be the unit ball of $\mathbb{R}^d$ under norm $||\cdot||_q$ for some $q \in [2, \infty)$, that is, $X = \{x \in \mathbb{R}^d : ||x||_q \leq 1\}$. Consider $\mathcal{F}$ consisting of a single rectified neuron $\mathcal{F} = \{\max(v^\top[x; 1], 0)^\alpha : v \in \mathbb{R}^{d+1}, ||v||_p = 1\}$ where $\alpha \in \mathbb{N}$, $\frac{1}{p} + \frac{1}{q} = 1$. Then we have*

$$d_{\mathrm{BL}}(\mu, \nu) = \tilde{O}(d_{\mathcal{F}}(\mu, \nu)^{\frac{1}{\alpha + (d+1)/2}}), \tag{6}$$

*where $\tilde{O}$ denotes the big-O notation ignoring the logarithm factor.*

The result in (6) shows that $d_{\mathcal{F}}(\mu, \nu)$ gives an increasingly weaker bound when the dimension $d$ increases. This is expected because we approximate a non-parametric set with a parametric one.

**Likelihood and KL divergence.** Maximum likelihood has been the predominant approach in statistical learning, and testing likelihood forms a standard criterion for testing unsupervised models. The recent advances in deep unsupervised learning, however, make it questionable whether likelihood is the right objective for training and evaluation (e.g., Theis et al., 2015). For example, some recent empirical studies (e.g., Danihelka et al., 2017; Grover et al., 2017) showed a counter-intuitive phenomenon that both the testing and training likelihood (assuming generators with valid densities are used) tend to decrease, instead of increase, as the GAN loss is minimized. A hypothesis for explaining this is that the neural distances used in GANs are too weak to control the KL divergence properly. Therefore, from the theoretical perspective, it is desirable to understand under what conditions (even if it is a very strong one), the neural distance can be strong enough to control KL divergence. This can be done by the following simple result.

**Proposition 2.9.** *Assume $\mu$ and $\nu$ have positive density functions $\rho_\mu(x)$ and $\rho_\nu(x)$, respectively. Then*

$$\mathrm{KL}(\mu||\nu) + \mathrm{KL}(\nu||\mu) = \mathbb{E}_\mu \log(\rho_\mu/\rho_\nu) - \mathbb{E}_\nu \log(\rho_\mu/\rho_\nu).$$

*If $\log(\rho_\mu/\rho_\nu) \in span\mathcal{F}$, then*

$$\mathrm{KL}(\mu||\nu) + \mathrm{KL}(\nu||\mu) \le || \log(\rho_\mu/\rho_\nu)||_{\mathcal{F},1} \, d_\mathcal{F}(\mu, \nu). \tag{7}$$

*If $\log(\rho_\mu/\rho_\nu) \in cl(span\mathcal{F})$ with an error decay function $\epsilon(r) = O(r^{-\kappa})$, then*

$$\mathrm{KL}(\mu||\nu) + \mathrm{KL}(\nu||\mu) = O(d_\mathcal{F}(\mu, \nu)^{\frac{\kappa}{\kappa+1}}). \tag{8}$$

This result shows that we require that the density ratio $\log(\rho_\mu/\rho_\nu)$ should exist and behave nicely in $span\mathcal{F}$ or $cl(span\mathcal{F})$ in order to bound KL divergence with $d_\mathcal{F}(\mu, \nu)$. If either $\mu$ or $\nu$ is an empirical measure, the bound is vacuum since $\mathrm{KL}(\mu, \nu) + \mathrm{KL}(\mu, \nu)$ equals infinite, while $d_\mathcal{F}(\mu, \nu)$ remains finite once $\mathcal{F}$ is bounded, i.e., $||f||_\infty \le \Delta < \infty$ for all $f \in \mathcal{F}$.

Obviously, this strong condition is hard to satisfy in practice, because practical data distributions and generators in GANs often have no densities or at least highly peaky densities. We draw more discussions in Corollary 3.5.

## 3 GENERALIZATION PROPERTY OF GANS

Section 2 suggests that it is better to use larger discriminator set $\mathcal{F}$ in order to obtain stronger neural distance. However, why do regularization techniques, which effectively shrink the discriminator set, help GAN training in practice? The answer has to do with the fact that we observe the true model $\mu$ only through an i.i.d. sample of size $m$ (whose empirical measure is denoted by $\hat{\mu}_m$), and hence can only optimize the empirical loss $d_\mathcal{F}(\hat{\mu}_m, \nu)$, instead of the exact loss $d_\mathcal{F}(\mu, \nu)$. Therefore, generalization bounds are required to control the exact loss $d_\mathcal{F}(\mu, \nu)$ when we can only minimize its empirical version $d_\mathcal{F}(\hat{\mu}_m, \nu)$. Specifically, let $\mathcal{G}$ be a class of generators that may or may not include the unknown true distribution $\mu$. Assume $\nu_m$ minimizes the GAN loss $d_\mathcal{F}(\hat{\mu}_m, \nu)$ up to an $\epsilon$ ($\epsilon \ge 0$) accuracy, that is,

$$d_\mathcal{F}(\hat{\mu}_m, \nu_m) \le \inf_{\nu \in \mathcal{G}} d_\mathcal{F}(\hat{\mu}_m, \nu) + \epsilon. \tag{9}$$

We are interested in bounding the difference between $\nu_m$ and the unknown $\mu$ under certain evaluation metric. Depending on what we care about, we may be interested in the generalization error in terms of the neural distance $d_\mathcal{F}(\mu, \nu_m)$, or other standard quantities of interest such as BL distance $d_{\mathrm{BL}}(\mu, \nu_m)$ and KL divergence $\mathrm{KL}(\mu, \nu_m)$ or the testing likelihood.

In this section, we adapt the standard Rademacher complexity argument to establish generalization bounds for GANs. We show that the discriminator set $\mathcal{F}$ should be small enough to be generalizable, striking a tradeoff with the other requirement that it should be large enough to be discriminative. We first present the generalization bound under neural distance, which purely depends on the Rademacher complexity of the discriminator set $\mathcal{F}$ and is independent of the generator set $\mathcal{G}$. Then using the results in Section (2.1), we discuss the generalization bounds under other standard metrics, like BL distance and KL divergence.

### 3.1 GENERALIZATION UNDER NEURAL DISTANCE

Using the standard derivation and the optimality condition (9), we have (see Appendix D)

$$d_{\mathcal{F}}(\mu, \nu_m) - \inf_{\nu \in \mathcal{G}} d_{\mathcal{F}}(\mu, \nu) \leq 2 \sup_{f \in \mathcal{F}} |\mathbb{E}_\mu[f] - \mathbb{E}_{\hat{\mu}_m}[f]| + \epsilon$$

$$= 2 d_{\mathcal{F}}(\mu, \hat{\mu}_m) + \epsilon. \tag{10}$$

This reduces the problem to bounding the discrepancy $d_{\mathcal{F}}(\mu, \hat{\mu}_m) := \sup_{f \in \mathcal{F}} |\mathbb{E}_\mu[f] - \mathbb{E}_{\hat{\mu}_m}[f]|$ between the true model $\mu$ and its empirical version $\hat{\mu}_m$. This can be achieved by the uniform concentration bounds developed in statistical learning theory (e.g., Vapnik & Vapnik, 1998) and empirical process (e.g., Van de Geer, 2000). In particular, the concentration property related to $\sup_{f \in \mathcal{F}} |\mathbb{E}_\mu[f] - \mathbb{E}_{\hat{\mu}_m}[f]|$ can be characterized by the Rademacher complexity of $\mathcal{F}$ (w.r.t. measure $\mu$), defined as

$$R_m^{(\mu)}(\mathcal{F}) := \mathbb{E}\left[\sup_{f \in \mathcal{F}} \frac{2}{m} \sum_i \tau_i f(X_i)\right], \tag{11}$$

where the expectation is taken w.r.t. $X_i \sim \mu$, and Rademacher random variable $\tau_i$: $\mathrm{prob}(\tau_i = 1) = \mathrm{prob}(\tau_i = -1) = 1/2$. Intuitively, $R_m^{(\mu)}(\mathcal{F})$ characterizes the ability of overfitting with pure random labels using functions in $\mathcal{F}$ and hence relates to the generalization bounds. Standard results in learning theory show that

$$\sup_{f \in \mathcal{F}} |\mathbb{E}_\mu[f] - \mathbb{E}_{\hat{\mu}_m}[f]| \leq R_m^{(\mu)}(\mathcal{F}) + O(\Delta \sqrt{\frac{\log(1/\delta)}{m}}),$$

where $\Delta = \sup_{f \in \mathcal{F}} ||f||_\infty$. Combining this with (10), we obtain the following result.

**Theorem 3.1.** *Assume that the discriminator set $\mathcal{F}$ is even, i.e., $f \in \mathcal{F}$ implies $-f \in \mathcal{F}$, and that all discriminators are bounded by $\Delta$, i.e., $||f||_\infty \leq \Delta$ for any $f \in \mathcal{F}$. Let $\hat{\mu}_m$ be an empirical measure of an i.i.d. sample of size $m$ drawn from $\mu$. Assume $\nu_m \in \mathcal{G}$ satisfies $d_{\mathcal{F}}(\hat{\mu}_m, \nu_m) \leq \inf_{\nu \in \mathcal{G}} d_{\mathcal{F}}(\hat{\mu}_m, \nu) + \epsilon$. Then with probability at least $1 - \delta$, we have*

$$d_{\mathcal{F}}(\mu, \nu_m) - \inf_{\nu \in \mathcal{G}} d_{\mathcal{F}}(\mu, \nu) \leq 2 R_m^{(\mu)}(\mathcal{F}) + 2\Delta \sqrt{\frac{2 \log(1/\delta)}{m}} + \epsilon, \tag{12}$$

*where $R_m^{(\mu)}(\mathcal{F})$ is the Rademacher complexity of $\mathcal{F}$ defined in (11).*

We obtain nearly the same generalization bound for neural $f$-divergence in Theorem B.3. Theorem 3.1 relates the generalization error of GANs to the Rademacher complexity of the discriminator set $\mathcal{F}$. The smaller the discriminator set $\mathcal{F}$ is, the more generalizable the result is. Therefore, the choice of $\mathcal{F}$ should strike a subtle balance between the generalizability and the discriminative power: $\mathcal{F}$ should be large enough to make $d_{\mathcal{F}}(\mu, \nu)$ discriminative as we discuss in Section 2.1, and simultaneously should be small enough to have a small generalization error in (12). It turns out parametric neural discriminators strike a good balance for this purpose, given that it is both discriminative as we show in Section 2.1, and give small generalization bound as we show in the following.

**Corollary 3.2.** *Let $X$ be the unit ball of $\mathbb{R}^d$ under norm $|| \cdot ||_2$, that is, $X = \{x \in \mathbb{R}^d : ||x||_2 \leq 1\}$. Assume that $\mathcal{F}$ is neural networks with a single rectified linear unit (ReLU) $\mathcal{F} = \{\max(v^\top[x; 1], 0) : v \in \mathbb{R}^{d+1}, ||v||_2 = 1\}$. Then with probability at least $1 - \delta$,*

$$d_{\mathcal{F}}(\mu, \nu_m) \leq \inf_{\nu \in \mathcal{G}} d_{\mathcal{F}}(\mu, \nu) + \frac{C}{\sqrt{m}} + \epsilon \tag{13}$$

*and*

$$d_{\mathrm{BL}}(\mu, \nu_m) = \tilde{O}\left(\left[\inf_{\nu \in \mathcal{G}} d_{\mathcal{F}}(\mu, \nu) + \frac{C}{\sqrt{m}} + \epsilon\right]^{\frac{1}{(d+3)/2}}\right), \tag{14}$$

*where $C = 4\sqrt{2} + 4\sqrt{\log(1/\delta)}$ and $\tilde{O}$ denotes the big-O notation ignoring the logarithm factor.*

Note that the three terms in Eqn. (13) take into account the modeling error ($\inf_{v \in \mathcal{G}} d_{\mathcal{F}}(\mu, \nu)$), sample complexity and generalization error ($C/\sqrt{m}$), and optimization error ($\epsilon$), respectively. Assuming zero modeling error and optimization error, we have (1) $d_{\mathcal{F}}(\mu, \nu_m) = O(m^{-1/2})$, which achieves the typical parametric convergence rate; (2) $d_{\mathrm{BL}}(\mu, \nu_m) = \tilde{O}(m^{-\frac{1}{d+3}})$, which becomes slower as the dimension $d$ increases. This decrease is because of the non-parametric nature of BL distance, instead of learning algorithm. As we show in Appendix A, we obtain a similar rate of $d_{\mathrm{BL}}(\mu, \nu_m) = O(m^{-\frac{1}{d}})$, even if we directly use BL distance as the learning objective.

Similar results can be obtained for general parametric discriminator class as follows.

**Corollary 3.3.** *Under the condition of Theorem 3.1, we further assume that (1) $\mathcal{F} = \{f_\theta : \theta \in \Theta \subset [-1, 1]^p\}$ is a parametric function class with $p$ parameters in a bounded set $\Theta$ and that (2) every $f_\theta$ is $L$-Lipschitz continuous with respect to the parameters $\theta$, i.e., $\|f_\theta - f_{\theta'}\|_\infty \leq L\|\theta - \theta'\|_2$. Then with probability at least $1 - \delta$, we have*

$$d_{\mathcal{F}}(\mu, \nu_m) \leq \inf_{\nu \in \mathcal{G}} d_{\mathcal{F}}(\mu, \nu) + \frac{C}{\sqrt{m}} + \epsilon, \tag{15}$$

*where $C = 16\sqrt{2\pi}pL + 2\Delta\sqrt{2\log(1/\delta)}$.*

This result can be easily applied to neural discriminators, since neural networks $f_\theta(x)$ are generally Lipschitz w.r.t. the parameter $\theta$, once the input domain $X$ is bounded. For neural discriminators, we also apply the bound on the Rademacher complexity of DNNs recently derived in Bartlett et al. (2017), which gives a sharper bound than that in Corollary 3.3; see Appendix A.1.

With the basic result in Theorem 3.1, we can also discuss the learning bounds of GANs with choices of non-parametric discriminators. Making use of Rademacher complexity of bounded sets in a RKHS (e.g., Lemma 22 in Bartlett & Mendelson (2003)), we give the learning bound of MMD-based GANs (Li et al., 2015; Dziugaite et al., 2015) as follows. We present the results for Wasserstein distance and total variance distance in Appendix A.2, and highlight the advantages of using parametric neural discriminators.

**Corollary 3.4.** *Under the condition of Theorem 3.1, we further assume that $\mathcal{F} = \{f \in \mathcal{H} : \|f\|_{\mathcal{H}} \leq 1\}$ where $\mathcal{H}$ is a RKHS whose positive definite kernel $k(x, x')$ satisfies $k(x, x) \leq C_k < +\infty$ for all $x \in X$. Then with probability at least $1 - \delta$,*

$$d_{\mathcal{F}}(\mu, \nu_m) \leq \inf_{\nu \in \mathcal{G}} d_{\mathcal{F}}(\mu, \nu) + \frac{C}{\sqrt{m}} + \epsilon, \tag{16}$$

*where $C = 2\left(2 + \sqrt{2\log(1/\delta)}\right)\sqrt{C_k}$.*

**Remark 3.1** (Comparisons with results in Arora et al. (2017)). *Arora et al. (2017) also discussed the generalization properties of GANs under a similar framework. In particular, they developed bounds of form $|d_{\mathcal{F}}(\mu, \nu) - d_{\mathcal{F}}(\hat{\mu}_m, \hat{\nu}_m)|$ where $\hat{\mu}_m$ and $\hat{\nu}_m$ are empirical versions of the target distribution $\mu$ and $\nu$ with sample size $m$. Our framework is similar, but considers bounding the quantity $d_{\mathcal{F}}(\mu, \nu_m) - \inf_{\nu \in \mathcal{G}} d_{\mathcal{F}}(\mu, \nu)$, which is of more direct interest. In fact, our Eqn. (10) shows that our generalization error can be bounded by the generalization error studied in Arora et al. (2017). Another difference is that we adapt the Rademacher complexity argument to derive the bound, while Arora et al. (2017) made use of the $\epsilon$-net argument.*

**Bounding the KL divergence and testing likelihood.** The above results depend on the evaluation metric we use, which is $d_{\mathcal{F}}(\mu, \nu)$ or $d_{\mathrm{BL}}(\mu, \nu)$. If we are interested in evaluating the model using even stronger metrics, such as KL divergence or equivalently testing likelihood, then the generator set $\mathcal{G}$ enters the scene in a more subtle way, in that a larger generator set $\mathcal{G}$ should be companioned with a larger discriminator set $\mathcal{F}$ in order to provide meaningful bounds on KL divergence. This is illustrated in the following result obtained by combining Theorem 3.1 and Proposition 2.9.

**Corollary 3.5.** *Assume both the true $\mu$ and all the generators $\nu \in \mathcal{G}$ have positive densities $\rho_\mu$ and $\rho_\nu$, respectively. Assume $\mathcal{F}$ consists of bounded functions with $\Delta := \sup_{f \in \mathcal{F}} \|f\|_\infty < \infty$.*

*Further, assume the discriminator set $\mathcal{F}$ is compatible with the generator set $\mathcal{G}$ in the sense that $\log(\rho_\nu/\rho_\mu) \in span\mathcal{F}, \forall \nu \in \mathcal{G}$, with a compatible coefficient defined as*

$$\Lambda_{\mathcal{F},\mathcal{G}} := \sup_{\nu \in \mathcal{G}} \|\log(\rho_\nu/\rho_\mu)\|_{\mathcal{F},1} < \infty.$$

*Then*

$$\mathrm{KL}(\mu, \nu_m) \leq \Lambda_{\mathcal{F},\mathcal{G}} \left( 2R_m^{(\mu)}(\mathcal{F}) + 2\Delta\sqrt{2\log(1/\delta)/m} + \Delta \inf_{\nu \in \mathcal{G}} \sqrt{\mathrm{KL}(\mu, \nu)} + \epsilon \right). \quad (17)$$

Different from the earlier bounds, the bound in (17) depends on the compatibility coefficient $\Lambda_{\mathcal{F},\mathcal{G}}$ that casts a more interesting trade-off on the choice of the generator set $\mathcal{G}$: the generator set $\mathcal{G}$ should be small and have well-behaved density functions to ensure a small $\Lambda_{\mathcal{F},\mathcal{G}}$, while should be large enough to have a small modeling error $\inf_{\nu \in \mathcal{G}} \sqrt{\mathrm{KL}(\mu, \nu)}$. Related, the discriminator set should be large enough to include all density ratios $\log(\rho_\mu/\rho_\nu)$ in a ball of radius $\Lambda_{\mathcal{F},\mathcal{G}}$ of $span\mathcal{F}$, and should also be small to have a low Rademacher complexity $R_m^{(\mu)}(\mathcal{F})$. Obviously, one can also extend Corollary 3.5 using (8) in Proposition 2.7, to allow $\log(\rho_\mu/\rho_\nu) \in cl(span\mathcal{F})$ in which case the compatibility of $\mathcal{G}$ and $\mathcal{F}$ should be mainly characterized by the error decay function $\epsilon(r)$.

$\mathrm{KL}(\mu, \nu_m) = \mathbb{E}_\mu[\log p_\mu] - \mathbb{E}_\mu[\log p_{\nu_m}]$ is the difference between the testing likelihood $\mathbb{E}_\mu[\log p_{\nu_m}]$ of estimated model $\nu_m$ and the optimal testing likelihood $\mathbb{E}_\mu[\log p_\mu]$. Therefore, Corollary 3.5 also provides a bound for testing likelihood. Unfortunately, the condition in Corollary 3.5 is rather strong, in that it requires that both the true distribution $\mu$ and the generators $\nu$ have positive densities and that the log-density ratio $\log(\rho_\mu/\rho_\nu)$ be well-behaved. In practical applications of computer vision, however, both $\mu$ and $\nu$ tend to concentrate on local regions or sub-manifolds of $X$, with very peaky densities, or even no valid densities; this causes the compatibility coefficient $\Lambda_{\mathcal{F},\mathcal{G}}$ very large, or infinite, making the bound in (17) loose or vacuum. This provides a potential explanation for some of the recent empirical findings (e.g., Danihelka et al., 2017; Grover et al., 2017) that the negative testing likelihood is uncorrelated with the GAN loss functions, or even increases during the GAN training progress. The underlying reason here is that the neural distance is not strong enough to provide meaningful bound for KL divergence. See Appendix E for an illustration using toy examples.

## 4 RELATED WORK

There is a surge of research interest in GANs; however, most of the work has been empirical in nature. There has been some theoretical literature on understanding GANs, including the discrimination and generalization properties of GANs.

The discriminative power of GANs is typically justified by assuming that the discriminator set $\mathcal{F}$ has enough capacity. For example, Goodfellow et al. (2014) assumes that $\mathcal{F}$ contains the optimal discriminator $\frac{p_{\mathrm{data}}(x)}{p_{\mathrm{data}}(x) + p_{\mathrm{g}}(x)}$. Similar capacity assumptions have been made in nearly all other GANs to prove their discriminative power; see, e.g., Zhao et al. (2016); Nowozin et al. (2016); Arjovsky et al. (2017). However, discriminators are in practice taken as certain parametric function class, like neural networks, which violates these capacity assumptions. The universal approximation property of neural networks is used to justify the discriminative power empirically. In this work, we show that the GAN loss is discriminative if $span\mathcal{F}$ can approximate any continuous functions. This condition is very weak and can be satisfied even when none of the discriminators is close to the optimal discriminator. The MMD-based GANs (Li et al., 2015; Dziugaite et al., 2015; Li et al., 2017a) avoid the parametrization of discriminators by taking advantage of the close-form solution of the optimal discriminator in the non-parametric RKHS space. Therefore, the capacity assumption is satisfied in MMD-based GANs, and their discriminative power is easily justified.

Liu et al. (2017) defines a notion of adversarial divergences that include a number of GAN objective functions. They show that if the objective function is an adversarial divergence with some additional conditions, then using a restricted discriminator family has a moment-matching effect. Our treatment of the neural divergence is directly inspired by them. We refer to Remark B.1 for a detailed comparison. Liu et al. (2017) also shows that for objective functions that are strict adversarial divergence, convergence in the objective function implies weak convergence. However, they do not provide a condition under which an adversarial divergence is strict. A major contribution of our work is to fill this gap, and to provide such a condition that is sufficient and necessary.

Dziugaite et al. (2015) studies generalization error, defined as $d_{\mathcal{F}}(\mu, \nu_m) - \inf_{\nu \in \mathcal{G}} d_{\mathcal{F}}(\mu, \nu)$ in our notation, for MMD-GAN in terms of fat-shattering dimension. Moreover, Dziugaite et al. (2015) obtains a generalization bound that incorporates the complexity of the hypothesis set $\mathcal{G}$. Although

their overall error bound is still $O(m^{-1/2})$, their work shows the possibility to sharpen our $\mathcal{G}$-independent bound. Arora et al. (2017) studies the generalization properties of GANs through the quantity $d_{\mathcal{F}}(\mu, \nu) - d_{\mathcal{F}}(\hat{\mu}_m, \hat{\nu}_m)$ (in our notations). The main difference between our work and Arora et al. (2017) is the definition of generalization error; see more discussions in Remark 3.1. Moreover, Arora et al. (2017) allows only polynomial number of samples from the generated distribution because the training algorithm should run in polynomial time. We do not consider this issue because in this work we only study the statistical properties of the objective functions and do not touch the optimization method. Finally, Arora et al. (2017) shows that the GAN loss can approach its optimal value even if the generated distribution has very low support, and (Arora & Zhang, 2017) provides empirical evidence for this problem. Our result is consistent with their results because our generalization error is measured by the neural distance/divergence.

Finally, there are some other lines of research on understanding GANs. Li et al. (2017b) studies the dynamics of GAN's training and finds that: a GAN with an optimal discriminator provably converges, while a first order approximation of the discriminator leads to unstable dynamics and mode collapse. Lei et al. (2017) studies WGAN and optimal transportation by convex geometry and provides a close-form formula for the optimal transportation map. Hu et al. (2017) provides a new formulation of GANs and variational autoencoders (VAEs), and thus unifies the most two popular methods to train deep generative models. We'd like to mention other recent interesting research on GANs, e.g., (Guo et al., 2017; Sinn & Rawat, 2017; Nock et al., 2017; Mescheder et al., 2017; Tolstikhin et al., 2017; Heusel et al., 2017).

## 5 CONCLUSIONS

We studied the discrimination and generalization properties of GANs with parameterized discriminator class such as neural networks. A neural distance is guaranteed to be discriminative whenever the linear span of its discriminator set is dense in the bounded continuous function space. On the other hand, a neural divergence is discriminative whenever the linear span of features defined by the last linear layer of its discriminators is dense in the bounded continuous function space. We also provided generalization bounds for GANs in different evaluation metrics. In terms of neural distance, our bounds show that generalization is guaranteed as long as the discriminator set is small enough, regardless of the size of the generator or hypothesis set. This raises an interesting discrimination-generalization balance in GANs. Fortunately, several GAN methods in practice already choose their discriminator set at the sweet point, where both the discrimination and generalization hold. Finally, our generalization bound in KL divergence provides an explanation on the counter-intuitive behaviors of testing likelihood in GAN training.

There are several directions that we would like to explore in the future. First of all, in this paper, we do not talk about methods to compute the neural distance/divergence. This is typically a non-concave maximization problem and is extremely difficult to solve. Many methods have been proposed to solve this kind of minimax problems, but both stable training methods and theoretical analysis of these algorithms are still missing. Secondly, our generalization bound depends purely on the discriminator set. It is possible to obtain sharper bounds by incorporating structural information from the generator set. Finally, we would like to extend our analysis to conditional GANs (see, e.g., Mirza & Osindero (2014); Springenberg (2015); Chen et al. (2016); Odena et al. (2016)), which have demonstrated impressive performance (Reed et al., 2016a;b; Zhang et al., 2017).

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

## A  GENERALIZATION ERROR OF OTHER DISCRIMINATOR SETS $\mathcal{F}$

### A.1  GENERALIZATION BOUND FOR NEURAL DISCRIMINATORS

For neural discriminators, we can use the following bound on the Rademacher complexity of DNNs, which was recently proposed in Bartlett et al. (2017).

**Theorem A.1.** *Let fixed activation functions $(\sigma_1, \ldots, \sigma_L)$ and reference matrices $(M_1, \ldots, M_L)$ be given, where $\sigma_i$ is $\rho_i$-Lipschitz and $\sigma_i(0) = 0$. Let spectral norm bounds $(s_1, \ldots, s_L)$ and matrix $(2,1)$ norm bounds $(b_1, \ldots, b_L)$ be given. Let $\mathcal{F}$ denote the discriminator set consisting of all choices of neural network $f_{\mathcal{A}}$:*

$$\mathcal{F}_{\mathrm{nn}} := \{f_{\mathcal{A}} \; : \; \mathcal{A} =: (A_1, \ldots, A_L), \|A_i\|_{\sigma} \leq s_i, \|A_i^T - M_i^T\|_{2,1} \leq b_i\}, \qquad (18)$$

*where $\|A\|_{\sigma} := \sigma_{\max}(A)$ and $\|A\|_{2,1} := \|(\|A_{:,1}\|_2, \ldots, \|A_{:,m}\|_2)\|_1$ are the matrix spectral norm and $(2,1)$ norm, respectively, and*

$$f_{\mathcal{A}}(x) := \sigma_L(A_L \sigma_{L-1}(A_{L-1} \ldots \sigma_1(A_1 x) \ldots))$$

*is the neural network associated with weight matrices $(A_1, \ldots, A_L)$. Moreover, assume that each matrix in $(A_1, \ldots, A_L)$ has dimension at most $W$ along each axis and define the spectral normalized complexity $R$ as*

$$R := \sqrt{\log(2W^2)} \left(\Pi_{j=1}^{L} s_j \rho_j\right) \left(\sum_{i=1}^{L} \left(\frac{b_i}{s_i}\right)^{2/3}\right)^{3/2}. \qquad (19)$$

*Let data matrix $X \in \mathbb{R}^{m \times d}$ be given, where the $m$ rows correspond to data points. When the sample size $m \geq 3\|X\|_F R$, the empirical Rademacher complexity satisfies*

$$\hat{R}_m(\mathcal{F}_{\mathrm{nn}}) := \mathbb{E}_{\boldsymbol{\tau}} \left[\sup_{f \in \mathcal{F}_{\mathrm{nn}}} \frac{2}{m} \sum_i \tau_i f(X_i)\right] \leq \frac{24\|X\|_F R}{m} \left(1 + \log \frac{m}{3\|X\|_F R}\right), \qquad (20)$$

*where $\boldsymbol{\tau} = (\tau_1, \ldots, \tau_m)$ are the Rademacher random variables which are iid with $Pr[\tau_i = 1] = Pr[\tau_i = -1] = 1/2$, $\|X\|_F$ is the Frobenius norm of $X$.*

*Proof.* The proof is the same with the proof of Lemma A.8 in Bartlett et al. (2017). When $m \geq 3\|X\|_F R$, we use the optimal $\alpha = 3\|X\|_F R/\sqrt{m}$ to obtain the above result. $\qquad \square$

Combined with our Theorem 3.1, we obtain the following generalization bound for the neural discriminator set defined in (18).

**Corollary A.2.** *Suppose that the discriminator set $\mathcal{F}_{\mathrm{nn}}$ is taken as (18) and that $\|f\|_{\infty} \leq \Delta$ for any $f \in \mathcal{F}_{\mathrm{nn}}$. Let data matrix $X \in \mathbb{R}^{m \times d}$ be the $m$ data points that define the empirical distribution $\hat{\mu}_m$. Then with probability at least $1 - \delta$, we have*

$$d_{\mathcal{F}_{\mathrm{nn}}}(\mu, \nu_m) \leq \inf_{\nu \in \mathcal{G}} d_{\mathcal{F}}(\mu, \nu) + \frac{48\|X\|_F R}{m} \left(1 + \log \frac{m}{3\|X\|_F R}\right) + 6\Delta\sqrt{\frac{2\log(2/\delta)}{m}} + \epsilon, \quad (21)$$

*where $R$ is the spectral normalized complexity defined in (19) and $\epsilon$ is the optimization error defined in (9).*

*Proof.* In the proof of Theorem 3.1, instead of using

$$\sup_{f \in \mathcal{F}} |\mathbb{E}_{\mu}[f] - \mathbb{E}_{\hat{\mu}_m}[f]| \leq R_m^{(\mu)}(\mathcal{F}) + 2\Delta\sqrt{\frac{\log(1/\delta)}{2m}},$$

we use

$$\sup_{f \in \mathcal{F}} |\mathbb{E}_{\mu}[f] - \mathbb{E}_{\hat{\mu}_m}[f]| \leq \hat{R}_m(\mathcal{F}) + 6\Delta\sqrt{\frac{\log(2/\delta)}{2m}}$$

to revise the generalization bound (12) as

$$d_{\mathcal{F}}(\mu, \nu_m) - \inf_{\nu \in \mathcal{G}} d_{\mathcal{F}}(\mu, \nu) \leq 2\hat{R}_m(\mathcal{F}) + 6\Delta\sqrt{\frac{2\log(1/\delta)}{m}} + \epsilon. \qquad (22)$$

Combining the revised bound with Equation (20), we conclude the proof. $\qquad \square$

Compared to Corollary 3.3, the bound in (22) gets rid of the number of parameters $p$, which can be prohibitively large in practice. Moreover, Corollary A.2 can be directly applied to the spectral normalized GANs (Anonymous, 2018), and may give an explanation of the empirical success of the spectral normalization technique.

## A.2 Generalization bounds for non-parametric discriminator sets

With the basic result in Theorem 3.1, we can also discuss the learning bounds of GANs with other choices of non-parametric discriminator sets $\mathcal{F}$. This allows us to highlight the advantages of using parametric neural discriminators. For simplicity, we assume zero model error and optimization so that the bound is solely based on the generalization error $d_{\mathcal{F}}(\mu, \hat{\mu}_m)$ between $\mu$ and its empirical version $\hat{\mu}_m$.

1. Bounded Lipschitz distance, $\mathcal{F} = \{f \in C(X) : \|f\|_{\mathrm{BL}} \leq 1\}$, which is equivalent to Wasserstein distance when $X$ is compact. When $X$ is a convex bounded set in $\mathbb{R}^d$, we have $R_m^{(\mu)}(\mathcal{F}) \leq m^{-1/d}$ for $d > 2$ (see Corollary 12 in Sriperumbudur et al. (2009)), and hence $d_{\mathrm{BL}}(\mu, \nu) = O(m^{-1/d})$. This is comparable with Corollary 3.2.

   This bound is tight. Assume that $\mu$ is the uniform distribution on $X$. A simple derivation (similar to Lemma 1 in Arora et al. (2017)) shows that $d_{\mathcal{F}}(\mu, \hat{\mu}_m) \geq c(1 - m \exp(-\Omega(d)))$ for some constant only depending on $X$. Therefore, one must need at least $m = \exp(\Omega(d))$ samples to reduce $d_{\mathcal{F}}(\mu, \hat{\mu}_m)$, and hence the generalization bound, to $O(\epsilon)$.

2. Total variation (TV) distance, $\mathcal{F} = \{f \in C(X) : \|f\| \leq \min\{1, \Delta\}\}$. It is easy to verify that $R_m^{(\mu)}(\mathcal{F}) = 2$. Therefore, Eqn. (12) cannot guarantee generalization even when we have infinite number of samples, i.e., $m \to \infty$.

   The estimate given in Eqn. (12) is tight. Assume that $\mu$ is the uniform distribution on $X$. It is easy to see that $d_{\mathrm{TV}}(\mu, \hat{\mu}_m) = 2$ almost surely. Therefore, $\nu_m$ is close to $\hat{\mu}_m$ implies that it is order 1 away from $\mu$, which means that generalization does not hold in this case.

With the statement that training with the TV distance does not generalize, we mean that training with TV distance does not generalize in TV distance. More precisely, even if the training loss on empirical samples is very small, i.e., $\mathrm{TV}(\hat{\mu}_m, \nu_m) = O(\epsilon)$, the TV distance to the unknown target distribution can be large, i.e., $d_{TV}(\mu, \nu_m) = O(1)$. However, this does not imply that training with TV distance is useless, because it is possible that training with a stronger metric leads to asymptotic vanishing in a weaker metric. For example, $d_{TV}(\hat{\mu}_m, \nu_m) = O(\epsilon)$ implies $d_{\mathcal{F}_{\mathrm{nn}}}(\hat{\mu}_m, \nu_m) = O(\epsilon)$, and thus a small $d_{\mathcal{F}_{\mathrm{nn}}}(\mu, \nu_m)$.

Take the Wasserstein metric as another example, even though we only establish $d_W(\mu, \nu_m) = O(m^{-1/d})$ (assuming zero model error ($\mu \in \mathcal{G}$) and optimization $\epsilon = 0$), it does not eliminate the possibility that the weaker neural distance has a faster convergence rate $d_{\mathcal{F}_{\mathrm{nn}}}(\mu, \nu_m) = O(m^{-1/2})$. From the practical perspective, however, TV and Wasserstein distances are less clearly favorable than neural distance because the difficulty of calculating and optimizing them.

## B Neural $\phi$-divergence

$f$-GAN is another broad family of GANs that are based on minimizing $f$-divergence (also called $\phi$-divergence) (Nowozin et al., 2016), which includes the original GAN by Goodfellow et al. (2014). [1] However, $\phi$-divergence has substantially different properties from IPM (see e.g., Sriperumbudur et al. (2009)), and is not defined as the intuitive moment matching form as IPM. In this Appendix, we extend our analysis to $\phi$-divergence by interpreting it as a form of *penalized moment matching*. Similar to the case of IPM, we analyze the *neural $\phi$-divergence* that restricts the discriminators to parametric function set $\mathcal{F}$ for practical computability, and establish its discrimination and generalization properties under mild conditions that practical $f$-GANs satisfy.

Assume that $\mu$ and $\nu$ are two distributions on $X$. Given a convex, lower-semicontinuous univariate function $\phi$ that satisfies $\phi(1) = 0$, the related $\phi$-divergence is $d_\phi(\mu \parallel \nu) = \mathbb{E}_\nu \left[\phi\left(\frac{\mathrm{d}\mu}{\mathrm{d}\nu}\right)\right]$. If $\phi$ is strictly convex, then a standard derivation based on Jensen's inequality shows that $\phi$-divergence is

---

[1]In this appendix, we call it $\phi$-divergence because $f$ has been used for discriminators.

nonnegative and discriminative: $d_\phi(\mu \,||\, \nu) \geq \phi(1) = 0$ and the equality holds iff $\mu = \nu$. Different choices of $\phi$ recover popular divergences as special cases. For example, $\phi(t) = (t-1)^2$ recovers Pearson $\chi^2$ divergence, and $\phi(t) = (u+1)\log((u+1)/2) + u \log u$ gives the Jensen-Shannon divergence used in the vanilla GAN Goodfellow et al. (2014).

## B.1 DISCRIMINATIVE POWER OF NEURAL $\phi$-DIVERGENCE

In this work, we find it helps to develop intuition by introducing another convex function $\psi(t) := \phi(t+1)$, defined by shifting the input variable of $\phi$ by $+1$; the $\phi$-divergence becomes

$$d_\phi(\mu \,||\, \nu) = \mathbb{E}_\nu \left[ \psi\left(\frac{\mathrm{d}\mu}{\mathrm{d}\nu} - 1\right) \right] = \int_X \rho_\nu(x) \psi\left(\frac{\rho_\mu(x)}{\rho_\nu(x)} - 1\right) \tau(\mathrm{d}x), \qquad (23)$$

where we should require that $\psi(0) = 0$; in right hand side of (23), we assume $\rho_\mu$ and $\rho_\nu$ are the density functions of $\mu$ and $\nu$, respectively, under a base measure $\tau$. The key advantage of introducing $\psi$ is that it gives a suggestive variational representation that can be viewed as a regularized moment matching. Specially, assume $\psi^*$ is the convex conjugate of $\psi$, that is, $\psi^*(t) = \sup_y\{yt - \psi(y)\}$. By standard derivation, we can show that

$$d_\phi(\mu \,||\, \nu) \geq \sup_{f \in \mathcal{A}} \left( \mathbb{E}_\mu[f] - \mathbb{E}_\nu[f] - \Psi_{\nu,\psi^*}[f] \right), \quad \text{with} \quad \Psi_{\nu,\psi^*}[f] := \mathbb{E}_{x \sim \nu}[\psi^*(f(x))], \quad (24)$$

where $\mathcal{A}$ is *the class of all functions* $f : X \to \mathrm{dom}(\psi^*)$ where $\mathrm{dom}(\psi^*) = \{t \colon \psi^*(t) \in \mathbb{R}\}$, and the equality holds if $\varphi^*(\frac{\rho_\mu(x)}{\rho_\nu(x)} - 1) \in \mathcal{A}$ where $\varphi$ is the inverse function of $\psi^{*\prime}$. In (24), the term $\Psi_{\nu,\psi^*}[f]$, as we show in Lemma B.1 in sequel, can be viewed as a type of complexity penalty on $f$ that ensures the supreme is finite. This is in contrast with the IPM $d_\mathcal{F}(\mu, \nu)$ in which the complexity constraint is directly imposed using the function class $\mathcal{F}$, instead of a regularization term.

**Lemma B.1.** *Assume* $\psi \colon \mathbb{R} \to \mathbb{R} \cup \{\infty\}$ *is a convex, lower-semicontinuous function with conjugate* $\psi^*$ *and* $\psi(0) = 0$. *The penalty* $\Psi_{\nu,\psi^*}[f]$ *in (24) has the following properties*

*i)* $\Psi_{\nu,\psi^*}[f]$ *is a convex functional of* $f$, *and* $\Psi_{\nu,\psi^*}[f] \geq 0$ *for any* $f$.

*ii) There exists a constant* $b_0 \in \mathbb{R} \cup \{\infty\}$ *such that* $\psi^*(b_0) = 0$. *Further, if* $\psi$ *is strictly convex, then* $\Psi_{\nu,\psi^*}[f] = 0$ *implies* $f(x) = b_0$ *almost surely under measure* $\nu$.

*Proof.* i) It is obvious that $\Psi_{\nu,\psi^*}[f]$ is convex given that $f^*$ is convex. By the convex conjugate, we have $\psi(t) = \sup_y \{ty - \psi^*(y)\}$. Take $t = 0$ and note that $\psi(0) = 0$, then we have $\psi^*(y) \geq 0$, $\forall y$. This proves $\Psi_{\nu,\psi^*}[f] \geq 0$.

ii) If $\psi$ is strictly convex, then $\psi^*$ is also strictly convex. This implies there exists at most a single value $b_0$ such that $\psi^*(c) = 0$. Given that $\psi^*(y) \geq 0$ for $\forall y$, we arrive that $\mathbb{E}_{x \sim \nu}[\psi^*(f(x))] = 0$ implies $\psi^*(f(x)) = 0$ almost surely under $x \sim \nu$, which then implies $f(x) = b_0$ almost surely. $\quad\square$

In practice, it is impossible to numerically optimize over the class of all functions in (24). Instead, practical $f$-GANs restrict the optimization to a parametric set $\mathcal{F}$ of neural networks, yielding the following *neural $\phi$-divergence*:

$$d_{\phi,\mathcal{F}}(\mu \,||\, \nu) = \sup_{f \in \mathcal{F}} \left( \mathbb{E}_\mu[f] - \mathbb{E}_\nu[f] - \Psi_{\nu,\psi^*}[f] \right). \qquad (25)$$

Note that this can be viewed as a generalization of the $\mathcal{F}$-related IPM $d_\mathcal{F}(\mu, \nu)$ by considering $\psi^* = 0$. However, the properties of the neural $\phi$-divergence can be significantly different from that of $d_\mathcal{F}(\mu, \nu)$. For example, $d_{\phi,\mathcal{F}}(\mu \,||\, \nu)$ is not even guaranteed to be non-negative for arbitrary discriminator sets $\mathcal{F}$ because of the negative regularization term. Fortunately, we can still establish the non-negativity and discriminative property of $d_{\phi,\mathcal{F}}(\mu \,||\, \nu)$ under certain weak conditions on $\mathcal{F}$. Moreover, the property that $d_\mathcal{F}(\mu, \nu) = 0$ implies moment matching on $\mathcal{F}$, which is the key step to establish the discriminative power, is not necessarily true for neural divergence. Fortunately, it turns out that $d_{\phi,\mathcal{F}}(\mu \,||\, \nu) = 0$ implies moment matching on features defined by the last linear layer of discriminators.

**Theorem B.1.** *Assume* $\mathcal{F}$ *includes the constant function* $b_0 \in \mathbb{R}$, *which satisfies* $\psi^*(b_0) = 0$ *as defined in Lemma B.1. We have*

*i)* $0 \leq d_{\phi,\mathcal{F}}(\mu \,||\, \nu) \leq d_{\mathcal{F}}(\mu, \nu)$ *for* $\forall \mu, \nu$*. As a result,*

$$d_{\mathcal{F}}(\mu, \nu) = 0 \quad \text{implies} \quad d_{\phi,\mathcal{F}}(\mu \,||\, \nu) = 0.$$

*In other words, moment matching on* $\mathcal{F}$ *is a sufficient condition of zero neural* $\phi$*-divergence.*

*ii) Further, we assume* $\mathcal{F}$ *has the following form:*

$$\mathcal{F} \supseteq \{\sigma(\alpha f_0 + c_0) \colon \forall |\alpha| \leq \alpha_{f_0}, \ \text{and} \ f_0 \in \mathcal{F}_0\}, \tag{26}$$

*where* $\mathcal{F}_0$ *is any function set, and* $\alpha_{f_0} > 0$ *is positive number associated with each* $f_0 \in \mathcal{F}_0$*, and* $c_0$ *is a constant and* $\sigma \colon \mathbb{R} \to \mathbb{R}$ *is any function that satisfies* $\sigma(c_0) = b_0$ *and* $\sigma'(c_0) > 0$*. Here* $\sigma$ *can be viewed as the output activation function of a deep neural network whose previous layers are specified by* $\mathcal{F}_0$*. Assume* $\psi^*(y)$ *is differentiable at* $y = b_0$*. Then*

$$d_{\phi,\mathcal{F}}(\mu \,||\, \nu) = 0 \quad \text{implies} \quad d_{\mathcal{F}_0}(\mu, \nu) = 0.$$

*In other words, moment matching on* $\mathcal{F}_0$ *is a necessary condition of zero neural* $\phi$*-divergence.*

*iii)* $cl(span\mathcal{F}_0) \supseteq C_b(X)$ *is a sufficient condition for* $d_{\phi,\mathcal{F}}$ *to be discriminative, i.e.,* $d_{\phi,\mathcal{F}}(\mu \,||\, \nu) = 0$ *implies* $\mu = \nu$*.*

Condition (26) defines a commonly used structure of $\mathcal{F}$ that naturally satisfied by the $f$-GANs used in practice; in particular, the output activation function $\sigma$ plays the role of ensuring the output of $\mathcal{F}$ respects the input domain of the convex function $\psi^*$. For example, the vanilla GAN has $\psi^* = -\log(1 - \exp(t)) - t$ with an input domain of $(-\infty, 0)$, and activation function is taken to be $\sigma(t) = -\log(1 + \exp(-t))$. See Table 2 of Nowozin et al. (2016) for the list of output activation functions related to commonly used $\psi$.

*Proof of Theorem B.1.* i) because $b_0 \in \mathcal{F}$ and $\psi^*(b_0) = 0$, we have $d_{\phi,\mathcal{F}}(\mu \,||\, \nu) \geq \mathbb{E}_\mu[b_0] - \mathbb{E}_\nu[b_0] - \Psi_{\nu,\psi^*}[b_0] = 0$. Because $\Psi_{\nu,\psi^*}[f] \geq 0$, we obtain $d_{\phi,\mathcal{F}}(\mu \,||\, \nu) \leq d_{\mathcal{F}}(\mu, \nu)$ by comparing (25) with $d_{\mathcal{F}}(\mu, \nu) = \sup_{f \in \mathcal{F}}\{\mathbb{E}_\mu f - \mathbb{E}_\nu f\}$.

ii), note that $d_{\psi,\mathcal{F}}(\mu \,||\, \nu) = 0$ implies $\mathbb{E}_\mu[f] - \mathbb{E}_\nu[f] \leq \Psi_{\nu,\psi^*}[f], \quad \forall f \in \mathcal{F}$. Therefore,

$$\mathbb{E}_\mu[\sigma(\alpha f_0 + c_0)] - \mathbb{E}_\nu[\sigma(\alpha f_0 + c_0)] \leq \Psi_{\nu,\psi^*}[\sigma(\alpha f_0 + c_0)], \quad \forall f_0 \in \mathcal{F}_0, \quad |\alpha| \leq \alpha_{f_0}.$$

This implies that

$$\frac{1}{\alpha}(\mathbb{E}_{x \sim \mu}[\sigma(\alpha f_0(x) + c_0)] - \mathbb{E}_{x \sim \nu}[\sigma(\alpha f_0(x) + c_0)]) \leq \frac{1}{\alpha}\mathbb{E}_{x \sim \nu}[\psi^*(\sigma(\alpha f_0(x) + c_0))]. \tag{27}$$

By the differentiability assumptions,

$$\lim_{\alpha \to 0} \frac{\sigma(\alpha f_0(x) + c_0) - \sigma(c_0)}{\alpha} = \sigma'(c_0)f(x),$$

$$\lim_{\alpha \to 0} \frac{\psi^*(\sigma(\alpha f_0(x) + c_0)) - \psi^*(\sigma(c_0))}{\alpha} = \psi^{*\prime}(b_0)\sigma'(c_0)f_0(x) = 0,$$

where we used the fact that $\psi^*(\sigma(c_0)) = \psi^*(b_0) = 0$ and $\psi^{*\prime}(b_0) = 0$ because $b_0$ is a differentiable minimum point of $\psi^*$. Taking the limit of $\alpha \to 0$ on both sides of (27), we get

$$\sigma'(c_0)[\mathbb{E}_{x \sim \mu}[f_0(x)] - \mathbb{E}_{x \sim \nu}[f_0(x)]] \leq 0, \quad \forall f_0 \in \mathcal{F}_0.$$

Because $\sigma'(c_0) > 0$ by assumption, this implies $\mathbb{E}_{x \sim \mu}[f_0(x)] - \mathbb{E}_{x \sim \nu}[f_0(x)]$. The same argument applies to $-f_0$, and we thus we finally obtain $\mathbb{E}_{x \sim \mu}[f_0(x)] = \mathbb{E}_{x \sim \nu}[f_0(x)]$.

iii) Combining Theorem 2.2 and the last point, we directly get the result. $\qquad\square$

**Remark B.1.** *Our results on neural* $\phi$*-divergence can in general extended to the more unified framework of Liu et al. (2017) in which divergences of form* $\max_f \mathbb{E}_{(x,y) \sim \mu \otimes \nu}[f(x,y)]$ *are studied. We choose to focus on* $\phi$*-divergence because of its practical importance. Our Theorem B.1 i) can be viewed as a special case of Theorem 4 of Liu et al. (2017) and our Theorem B.1 ii) is related to Theorem 5 of Liu et al. (2017). However, Theorem 5 of Liu et al. (2017) requires a rather counterintuitive condition, while our condition in Theorem B.1 ii) is clear and satisfied by all* $\phi$*-divergence listed in Nowozin et al. (2016).*

Similar to Theorem 2.5, under the conditions of Theorem B.1, we have similar results for neural $\phi$-divergence.

**Theorem B.2.** *Using the same notions in Theorem B.1, assume that $(X, d_X)$ is a compact metric space and that $cl(span\mathcal{F}_0) \supseteq C_b(X)$. Then if $\lim_{n \to \infty} d_{\phi,\mathcal{F}}(\mu \,||\, \nu_n) = 0$, $\nu_n$ converges to $\mu$ in the distribution sense.*

*Further, if there exists $C > 0$ such that $\mathcal{F} \subset \{f \in C(X) : \|f\|_{Lip} \leq C\}$, we have*

$$\lim_{n \to \infty} d_{\phi,\mathcal{F}}(\mu \,||\, \nu_n) = 0 \Longleftrightarrow \nu_n \rightharpoonup \mu.$$

Notice that we assume that $(X, d_X)$ be a compact metric space here for simplicity. A non-compact result is available but its proof is messy and non-intuitive.

*Proof.* The first half is a direct application of Theorem B.1 and Theorem 10 in Liu et al. (2017).

For the second half, we have

$$d_{\phi,\mathcal{F}}(\mu \,||\, \nu_n) \leq d_{\mathcal{F}}(\mu, \nu_n) \leq C d_W(\mu, \nu_n),$$

where we use Theorem B.1 i) in the first inequality and the Lipschitz condition of $\mathcal{F}$ in the second ineqaulity. Since $d_W$ metrizes the weak convergence for compact $X$, we obtain $d_W(\mu, \nu_n) \to 0$ and thus $d_{\phi,\mathcal{F}}(\mu \,||\, \nu_n) \to 0$. $\qquad\square$

### B.2 GENERALIZATION PROPERTIES OF NEURAL $\phi$-DIVERGENCE

Similar to the case of neural distance, we can establish generalization bounds for neural $\phi$-divergence.

**Theorem B.3.** *Assume that $\|f\|_\infty \leq \Delta$ for any $f \in \mathcal{F}$. $\hat{\mu}_m$ is an empirical distribution with $m$ samples from $\mu$, and $\nu_m \in \mathcal{G}$ satisfies $d_{\phi,\mathcal{F}}(\hat{\mu}_m \,||\, \nu_m) \leq \inf_{\nu \in \mathcal{G}} d_{\phi,\mathcal{F}}(\hat{\mu}_m \,||\, \nu) + \epsilon$. Then with probability at least $1 - 2\delta$, we have*

$$d_{\phi,\mathcal{F}}(\mu \,||\, \nu_m) \leq \inf_{\nu \in \mathcal{G}} d_{\phi,\mathcal{F}}(\mu \,||\, \nu) + 2R_m^{(\mu)}(\mathcal{F}) + 2\Delta\sqrt{\frac{2\log(1/\delta)}{m}} + \epsilon, \qquad (28)$$

*where $R_m^{(\mu)}(\mathcal{F})$ is the Rademacher complexity of $\mathcal{F}$.*

Notice that the only difference between Theorem 3.1 and Theorem B.3 is that the failure probability change from $\delta$ to $2\delta$. This comes from the fact that $\mathcal{F}$ is typically not even in the neural divergence case. For example, the vanilla GAN takes $\sigma(t) = -\log(1 + \exp(-t))$ as the output activation function, and thus $f \leq 0$ for all $f \in \mathcal{F}$.

*Proof of Theorem B.3.* With the same argument in Equation (10), we obtain

$$d_{\phi,\mathcal{F}}(\mu \,||\, \nu_m) - \inf_{\nu \in \mathcal{G}} d_{\phi,\mathcal{F}}(\mu \,||\, \nu) \leq 2\sup_{f \in \mathcal{F}} |\mathbb{E}_\mu[f] - \mathbb{E}_{\hat{\mu}_m}[f]| + \epsilon.$$

Although $\mathcal{F}$ is not even, we have

$$\sup_{f \in \mathcal{F}} |\mathbb{E}_\mu[f] - \mathbb{E}_{\hat{\mu}_m}[f]| = \max\left\{ \sup_{f \in \mathcal{F}} \left(\mathbb{E}_\mu[f] - \mathbb{E}_{\hat{\mu}_m}[f]\right), \sup_{f \in \mathcal{F}} \left(\mathbb{E}_{\hat{\mu}_m}[f] - \mathbb{E}_\mu[f]\right) \right\}.$$

Standard argument on Rademacher complexity (same in the proof of Theorem 3.1) gives with probalibity at least $1 - \delta$,

$$\mathbb{E}\left[ \sup_{f \in \mathcal{F}} \left(\mathbb{E}_\mu[f] - \mathbb{E}_{\hat{\mu}_m}[f]\right) \right] \leq R_m(\mathcal{F}) + 2\Delta\sqrt{\frac{\log(1/\delta)}{2m}}.$$

With the same argument, we obtain that with probalibity at least $1 - \delta$,

$$\mathbb{E}\left[ \sup_{f \in \mathcal{F}} \left(\mathbb{E}_{\hat{\mu}_m}[f] - \mathbb{E}_\mu[f]\right) \right] \leq R_m(\mathcal{F}) + 2\Delta\sqrt{\frac{\log(1/\delta)}{2m}}.$$

Combining all the results above, we conclude the proof. $\qquad\square$

With Theorem B.3, we obtain generalization bounds for difference choices of $\mathcal{F}$, as we had in section 3. For example, we have an analog of Corollary 3.3 in the neural divergence setting as follows.

**Corollary B.4.** *Under the condition of Theorem B.3, we further assume that (1) $\mathcal{F} = \mathcal{F}_{nn} = \{f_\theta : \theta \in \Theta \subset [-1,1]^p\}$ is a parametric function class with $p$ parameters in a bounded set $\Theta$ and that (2) every $f_\theta$ is $L$-Lipschitz continuous with respect to the parameters $\theta$. Then with probability at least $1 - 2\delta$, we have*

$$d_{\phi,\mathcal{F}_{nn}}(\mu \parallel \nu_m) \leq \inf_{\nu \in \mathcal{G}} d_{\phi,\mathcal{F}_{nn}}(\mu \parallel \nu) + \frac{C}{\sqrt{m}} + \epsilon, \tag{29}$$

*where $C = 16\sqrt{2\pi}pL + 2\Delta\sqrt{2\log(1/\delta)}$.*

## C  PROOF OF RESULTS IN SECTION 2

*Proof of Theorem 2.2.* For the sufficient part, the proof is standard and the same as that of the uniqueness of weak convergence. We refer to Lemma 9.3.2 in Dudley (2002) for a complete proof.

For the necessary part, suppose that $\mathcal{F} \subset C_b(X)$ is discriminative in $\mathcal{P}_\mathcal{B}(X)$. Assume that $cl(\text{span}(\mathcal{F} \cup \{\mathbf{1}\}))$ is a strictly closed subspace of $C_b(X)$. Take $g \in C_b(X)\backslash cl(\text{span}(\mathcal{F}))$ and $\|g\|_\infty = 1$. By the Hahn-Banach theorem, there exists a bounded linear functional $L : C(X) \to \mathbb{R}$ such that $L(f) = 0$ for any $f \in cl(\text{span}(\mathcal{F} \cup \{\mathbf{1}\}))$ and $L \neq \mathbf{0}$. Thanks to the Riesz representation theorem for compact metric spaces, there exists a signed, regular Borel measure $m \in M_\mathcal{B}(X)$ such that

$$L(f) = \int_m f \quad \forall f \in C_b(X).$$

Suppose $m = \mu - \nu$ are the Hahn decomposition of $m$, where $\mu$ and $\nu$ are two nonnegative Borel measures. Then we have $L(f) = \int_\mu f - \int_\nu f$ for any $f \in C_b(X)$. Thanks to $L(\mathbf{1}) = 0$, we have $0 < \mu(X) = \nu(X) < \infty$. We can assume that $\mu$ and $\nu$ are Borel probability measures. (Otherwise, we can use the normalized nonzero linear functional $L/\mu(X)$ whose Hahn decomposition consists of two Borel probability measures.) Since $L(f) = 0$ for any $f \in cl(\text{span}(\mathcal{F}))$, we have $\int_\mu f = \int_\nu f$ for any $f \in \mathcal{F}$. Since $\mathcal{F} \subset C_b(X)$ is discriminative, we have $\mu = \nu$ and thus $L = \mathbf{0}$, which leads to a contradiction. $\square$

*Proof of Corollary 2.4.* Thanks to $\{\lambda\theta : \lambda \geq 0, \theta \in \Theta\} = \mathbb{R}^{d+1}$, for any $[w, b] \in \mathbb{R}^{n+1}$, there exists $[w_0, b_0] \in \Theta$ and $\lambda > 0$ such that

$$\sigma(w^\top x + b) = \sigma(\lambda(w_0^\top x + b_0)) = \lambda^\alpha \sigma(w_0^\top x + b_0),$$

where we used $\sigma(u) = \max\{u, 0\}^\alpha$ in the last step. Therefore, we have

$$span\mathcal{F}_{nn} \subset span\{\sigma(w^\top x + b) \colon [w, b] \in \mathbb{R}^{d+1}\}.$$

Thanks to Theorem 2.3, we know that $span\mathcal{F}_{nn}$ is dense in $C_b(X)$. $\square$

*Proof of Theorem 2.5.* Given a function $g \in C_b(X)$, we say that $g$ is approximated by $\mathcal{F}$ with error decay function $\epsilon(r)$ if for any $r \geq 0$, there exists $f_r \in span\mathcal{F}$ with $\|f_r\|_{\mathcal{F},1} \leq r$ such that $\|f - f_r\|_\infty \leq \epsilon(r)$. Obviously, $\epsilon(r)$ is an non-increasing function w.r.t. $r$. Thanks to $cl(span\mathcal{F}) = C_b(X)$, we have $\lim_{r\to\infty} \epsilon(r) = 0$. Now denote $r_n := d_\mathcal{F}(\mu, \nu_n)^{-1/2}$ and correspondingly $f_n := f_{r_n}$. We have

$$\begin{aligned}
|\mathbb{E}_\mu g - \mathbb{E}_{\nu_n} g| &\leq |\mathbb{E}_\mu g - \mathbb{E}_\mu f_n| + |\mathbb{E}_\nu g - \mathbb{E}_\nu f_n| + |\mathbb{E}_\mu f_n - \mathbb{E}_{\nu_n} f_n| \\
&\leq 2\epsilon(r_n) + r_n\, d_\mathcal{F}(\mu, \nu_n). = 2\epsilon(r_n) + 1/r_n.
\end{aligned}$$

If $\lim_{n\to\infty} d_\mathcal{F}(\mu, \nu_n) = 0$, we have $\lim_{n\to\infty} r_n = \infty$. Thanks to $\lim_{r\to\infty} \epsilon(r) = 0$, we prove that $\lim_{n\to\infty} |\mathbb{E}_\mu g - \mathbb{E}_{\nu_n} g| = 0$. Since this holds true for any $g \in C_b(X)$, we conclude that $\nu_n$ weakly converges to $\mu$.

If $\mathcal{F} \subseteq \text{BL}_C(X)$ for some $C > 0$, we have $d_\mathcal{F}(\mu, \nu) \leq Cd_{BL}(\mu, \nu)$ for any $\mu, \nu$. Because the bounded Lipschitz distance (also called FortetMourier distance) metrizes the weak convergence, we obtain that $\nu_n \rightharpoonup \mu$ implies $d_{BL}(\mu, \nu_n) \to 0$, and thus $d_\mathcal{F}(\mu, \nu_n) \to 0$. $\square$

*Proof of Proposition 2.6.* Let $g = \sum_{i=1}^{n} w_i f_i + w_0$. Then we have

$$| \mathbb{E}_\mu \, g - \mathbb{E}_\nu \, g | = \sum_{i=1}^{n} |w_i|| \mathbb{E}_\mu \, f_i - \mathbb{E}_\nu \, f_i | \leq \left( \sum_{i=1}^{n} |w_i| \right) d_{\mathcal{F}}(\mu, \nu).$$

The result is obtain by taking infimum over all possible $w_i$. □

*Proof of Proposition 2.7.* For any $r \geq 0$, we have

$$| \mathbb{E}_\mu \, g - \mathbb{E}_\nu \, g | \leq | \mathbb{E}_\mu \, g - \mathbb{E}_\mu \, f_r | + | \mathbb{E}_\nu \, g - \mathbb{E}_\nu \, f_r | + | \mathbb{E}_\mu \, f_r - \mathbb{E}_\nu \, f_r | \leq 2\epsilon(r) + r \, d_{\mathcal{F}}(\mu, \nu).$$

Taking the infimum on $r > 0$ on the right side gives the result. □

*Proof of Corollary 2.8.* Proposition 5 of Bach (2017) shows that for any bounded Lipschitz function $g$ that satisfies $||g||_{\mathrm{BL}} := \max\{||g||_\infty, ||g||_{\mathrm{Lip}}\} \leq \eta$, we have $\epsilon(r) = O(\eta(r/\eta)^{-1/(\alpha+(d-1)/2)} \log(r/\eta))$. Using Proposition 2.7, we get

$$| \mathbb{E}_\mu \, g - \mathbb{E}_\nu \, g | \leq \tilde{O}(||g||_{\mathrm{BL}} \, d_{\mathcal{F}}(\mu, \nu)^{\frac{1}{\alpha+(d+1)/2}}),$$

The result follows $\mathrm{BL}(\mu, \nu) = \sup_g \{| \mathbb{E}_\mu \, g - \mathbb{E}_\nu \, g | : \; ||g||_{\mathrm{BL}} \leq 1\}$. □

# D   PROOF OF RESULTS IN SECTION 3

**Proof of Equation** (10)   Using the standard derivation and the optimality condition (9), we have

$$d_{\mathcal{F}}(\mu, \nu_m) - \inf_{\nu \in \mathcal{G}} d_{\mathcal{F}}(\mu, \nu)$$
$$= d_{\mathcal{F}}(\mu, \nu_m) - d_{\mathcal{F}}(\hat{\mu}_m, \nu_m) + d_{\mathcal{F}}(\hat{\mu}_m, \nu_m) - \inf_{\nu \in \mathcal{G}} d_{\mathcal{F}}(\mu, \nu)$$
$$\leq d_{\mathcal{F}}(\mu, \nu_m) - d_{\mathcal{F}}(\hat{\mu}_m, \nu_m) + \inf_{\nu \in \mathcal{G}} d_{\mathcal{F}}(\hat{\mu}_m, \nu) - \inf_{\nu \in \mathcal{G}} d_{\mathcal{F}}(\mu, \nu) + \epsilon.$$

Therefore, we obtain

$$d_{\mathcal{F}}(\mu, \nu_m) - \inf_{\nu \in \mathcal{G}} d_{\mathcal{F}}(\mu, \nu) \leq 2 \sup_{\nu \in \mathcal{G}} |d_{\mathcal{F}}(\mu, \nu) - d_{\mathcal{F}}(\hat{\mu}_m, \nu)| + \epsilon.$$

Combining with the definition (1), we obtain

$$d_{\mathcal{F}}(\mu, \nu_m) - \inf_{\nu \in \mathcal{G}} d_{\mathcal{F}}(\mu, \nu) \leq 2 \sup_{f \in \mathcal{F}} |\mathbb{E}_\mu[f] - \mathbb{E}_{\hat{\mu}_m}[f]| + \epsilon.$$

*Proof of Theorem 3.1.* First of all, since $\mathcal{F}$ is even, we have $\sup_{f \in \mathcal{F}} |\mathbb{E}_\mu[f] - \mathbb{E}_{\hat{\mu}_m}[f]| = \sup_{f \in \mathcal{F}} (\mathbb{E}_\mu[f] - \mathbb{E}_{\hat{\mu}_m}[f])$. Consider the function

$$h(X_1, X_2, \ldots, X_m) = \sup_{f \in \mathcal{F}} (\mathbb{E}_\mu[f] - \mathbb{E}_{\hat{\mu}_m}[f]).$$

Since $f$ takes values in $[-\Delta, \Delta]$, changing $X_i$ to another independent copy $X_i'$ can change $h$ by no more than $2\Delta/m$. McDiarmid's inequality implies that with probability at least $1 - \delta$,

$$\sup_{f \in \mathcal{F}} (\mathbb{E}_\mu[f] - \mathbb{E}_{\hat{\mu}_m}[f]) \leq \mathbb{E} \left[ \sup_{f \in \mathcal{F}} (\mathbb{E}_\mu[f] - \mathbb{E}_{\hat{\mu}_m}[f]) \right] + 2\Delta \sqrt{\frac{\log(1/\delta)}{2m}}.$$

Standard argument on Rademacher complexity gives

$$\mathbb{E} \left[ \sup_{f \in \mathcal{F}} (\mathbb{E}_\mu[f] - \mathbb{E}_{\hat{\mu}_m}[f]) \right] \leq 2 \mathbb{E}_{\boldsymbol{\tau}, \boldsymbol{X}} \left[ \sup_{f \in \mathcal{F}} \frac{1}{m} \sum_i \tau_i f(X_i) \right] := R_m(\mathcal{F}).$$

Combining the two estimates above and Eqn. (10), we conclude the proof. □

*Proof of Corollary 3.2.* Part of the proof is from Proposition 7 in Bach (2017). More accurately, the discriminator set we use here is

$$\mathcal{F} = \left\{ \sum_{i=1}^{n} w_i \max(v_i^\top [x; 1], 0) \colon \sum_{i=1}^{n} |w_i| \leq 1, \quad \|v_i\|_2 = 1 \forall 1 \leq i \leq n \right\}$$

for a fix $n \in \mathbb{N}$. Since $\|x\|_2 \leq 1$ and $\|v\|_2 \leq 1$, it is easy to see that $\|f\|_\infty \leq \sqrt{2}$ for all $f \in \mathcal{F}$.

We want to estimate $R_m^{(\mu)}(\mathcal{F})$ and then use Theorem 3.1 to prove the result. First, it's easy to verify that

$$\sup_{f \in \mathcal{F}} \left| \frac{2}{m} \sum_i \tau_i f(X_i) \right| = \sup_{\|v\|_2 = 1} \left| \frac{2}{m} \sum_i \tau_i \max(v^\top [X_i; 1], 0) \right|.$$

Then we have

$$R_m^{(\mu)}(\mathcal{F}) = \mathbb{E}\left[ \sup_{\|v\|_2 = 1} \left| \frac{2}{m} \sum_i \tau_i \max(v^\top [X_i; 1], 0) \right| \right]$$

$$\leq \mathbb{E}\left[ \sup_{\|v\|_2 = 1} \left| \frac{2}{m} \sum_i \tau_i v^\top [X_i; 1] \right| \right] = \frac{2}{m} \mathbb{E}\left[ \left\| \sum_i \tau_i [X_i; 1] \right\|_2 \right],$$

where we use the 1-Lipschitz property of $\max(x, 0)$ and Talagrand's contraction lemma (Ledoux & Talagrand, 2013) in the inequality step. From Kakade et al. (2009), we get the Rademacher complexity of linear functions

$$\mathbb{E}\left[ \left\| \sum_i \tau_i [X_i; 1] \right\|_2 \right] \leq \sqrt{2m}.$$

Therefore, we obtain

$$R_m^{(\mu)}(\mathcal{F}) \leq \frac{2\sqrt{2}}{\sqrt{m}}.$$

Combined with $\|f\|_\infty \leq \sqrt{2}$ and Theorem 3.1, we finish the proof. $\qquad \square$

*Proof of Corollary 3.3.* We need to derive a bound for the Rademacher complexity (11). For fixed $\{x_i\}_{i=1}^m$, let's consider $X_\theta = \frac{1}{L\sqrt{m}} \sum_{i=1}^m \tau_i f_\theta(x_i)$. First of all, $\{X_\theta : \theta \in \Theta\}$ is a sub-Gaussian process with respect to the Eulidean distance on $\Theta$, i.e., for all $\theta, \theta' \in \Theta$ and all $\lambda > 0$

$$\mathbb{E}\left[ \exp\left( \lambda(X_\theta - X_{\theta'}) \right) \right] \leq \exp\left( \frac{\lambda^2 \|\theta - \theta'\|_2^2}{2} \right).$$

This is a standard result and can be derived by the Hoeffding's lemma. Secondly, we have the following bound for the $\epsilon$-cover number of $\Theta$:

$$\log N(\epsilon, \Theta, \|\cdot\|_2) \leq p \log(\sqrt{p}/\epsilon). \tag{30}$$

This bound is from the following simple construction. Consider a uniform grid with grid size $2\epsilon/\sqrt{p}$. The balls with centers as the grid points and with radius $\epsilon$ cover the unit cubic on $\mathbb{R}^p$, i.e., $[-1, 1]^p$. The number of balls in this construction is $(\sqrt{p}/\epsilon)^p$. Finally, by applying Dudley's entropy integral, we have

$$\mathbb{E}\left[ \sup_{\theta \in \Theta} X_\theta \right] \leq 8\sqrt{2} \int_0^{\sqrt{p}} \sqrt{\log N(\epsilon, \Theta, \|\cdot\|_2)} d\epsilon \tag{31}$$

Combing (30) and (31) and taking $\tau = \log(\sqrt{p}/\epsilon)$, we obtain

$$\mathbb{E}\left[ \sup_{\theta \in \Theta} X_\theta \right] \leq 8\sqrt{2} p \int_0^\infty \tau^{1/2} e^{-\tau} d\tau = 4\sqrt{2\pi} p. \tag{32}$$

Notice that Eqn. (32) holds true for arbitrary samples $\{x_i\}_{i=1}^m$, and thus we conclude that

$$R_m^{(\mu)}(\mathcal{F}) \leq C/\sqrt{m},$$

where $C = 8\sqrt{2\pi} pL$. $\qquad \square$

*Proof of Corollary 3.4.* Lemma 22 in Bartlett & Mendelson (2003) shows that if $\sup_{x \in X} k(x,x) \leq C_k \leq +\infty$, we have $R_m^{(\mu)}(\mathcal{F}) \leq 2\sqrt{C_k/m}$ for any $\mu \in \mathcal{P}_\mathcal{B}(X)$. Also, note that $f(x) \leq ||f||_H \sqrt{k(x,x)} \leq ||f||_H \sqrt{C_k}$. Combined with Theorem 3.1, we conclude the proof. $\qquad\square$

*Proof of Corollary 3.5.* Use Proposition 3.1 and note that $\mathrm{KL}(\mu, \nu_m) \leq \Lambda_{\mathcal{F},\mathcal{G}} \, d_\mathcal{F}(\mu,\nu)$ and $d_\mathcal{F}(\mu,\nu) \leq \Delta \mathrm{TV}(\mu,\nu) \leq \Delta\sqrt{2\mathrm{KL}(\mu,\nu)}$ by Pinsker's inequality. $\qquad\square$

# E  INCONSISTENCY BETWEEN GAN'S LOSS AND TESTING LIKELIHOOD

In this section, we will test our analysis of the consistency of GAN objective and likelihood objective on two toy datasets, e.g., a 2D Gaussian dataset and a 2D 8-Gaussian mixture dataset.

## E.1  A 2D GAUSSIAN EXAMPLE

The underlying ground-truth distribution is a 2D Gaussian with mean $(0.5, -0.5)$ and covariance matrix $\frac{1}{128}\begin{bmatrix} 17 & 15 \\ 15 & 17 \end{bmatrix}$. We take $10^5$ samples for training, and 1000 samples for testing.

For a 2D Gaussian distribution, we use the following generator

$$\begin{bmatrix} x_1 \\ x_2 \end{bmatrix} = \begin{bmatrix} 1 & \\ l & 1 \end{bmatrix} \begin{bmatrix} e^{s_1} & \\ & e^{s_2} \end{bmatrix} \begin{bmatrix} z_1 \\ z_2 \end{bmatrix} + \begin{bmatrix} b_1 \\ b_2 \end{bmatrix}, \tag{33}$$

where $\boldsymbol{z} = \begin{bmatrix} z_1 \\ z_2 \end{bmatrix}$ is a standard 2D normal random vector, and $l \in \mathbb{R}$, $\boldsymbol{s} = \begin{bmatrix} s_1 \\ s_2 \end{bmatrix} \in \mathbb{R}^2$ and $\boldsymbol{b} = \begin{bmatrix} b_1 \\ b_2 \end{bmatrix} \in \mathbb{R}^2$ are trainable parameters in the generator.

We train the generative model by WGAN with weight clipping. In the first experiment, the discriminator set is a neural network with one hidden layer and 500 hidden neurons, i.e.,

$$\mathcal{F}_{\text{nn}} = \{\sum_{i=1}^{500} \alpha_i \max(\boldsymbol{w}_i^\top [x; 1], \, 0) : -0.05 \leq \boldsymbol{\alpha} \leq 0.05, -0.05 \leq \boldsymbol{w}_i \leq 0.05 \quad \forall i\}.$$

Motivated by Corollary 3.5, in the second experiment, we take the discriminators to be the log-density ratio between two Gaussian distributions, which are quadratic polynomials:

$$\mathcal{F}_{\text{quad}} = \{x^\top A x + \boldsymbol{b}^\top x : -0.05 \leq A \leq 0.05, -0.05 \leq \boldsymbol{b} \leq 0.05\}.$$

We plot their results in Figure 1. We can see that both discriminators behave well: The training loss (the neural distance) converge to zero, and the testing log likelihood increases monotonically during the training. However, the quadratic polynomial discriminators $\mathcal{F}_{\text{quad}}$ yields higher testing log likelihood and better generative model at the convergence. This is expected because Corollary 3.5 guarantees that the testing log likelihood is bounded by the GAN loss (up to a constant), while it is not true for $\mathcal{F}_{\text{nn}}$.

We can also maximize the likelihood (MLE) on the training dataset to train the model, and we show its result in Figure 3. We can see that both MLE and Q-GAN (refers to WGAN with the quadratic discriminator $\mathcal{F}_{\text{quad}}$) yield similar results. However, directly maximizing the likelihood converges much faster than the WGAN in this example.

In this simple Gaussian example, the WGAN loss and the testing log likelihood are consistent. We indeed observe that by carefully choosing the discriminator set (as suggested in Corollary 3.5), the testing log likelihood can be simultaneously optimized as we optimize the GAN objective.

## E.2  AN EXAMPLE OF 2D 8-GAUSSIAN MIXTURE

The underlying ground truth distribution is a 2D Gaussian mixture with 8 Gaussians and with equal weights. Their centers are distributed equally on the circle centered at the origin and with radius

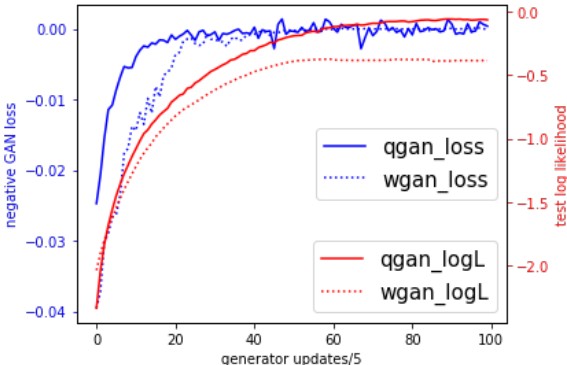

Figure 1: Negative GAN losses and testing likelihood. qgan: WGAN with quadratic polynomials as discriminator. wgan: WGAN with neural discriminator.

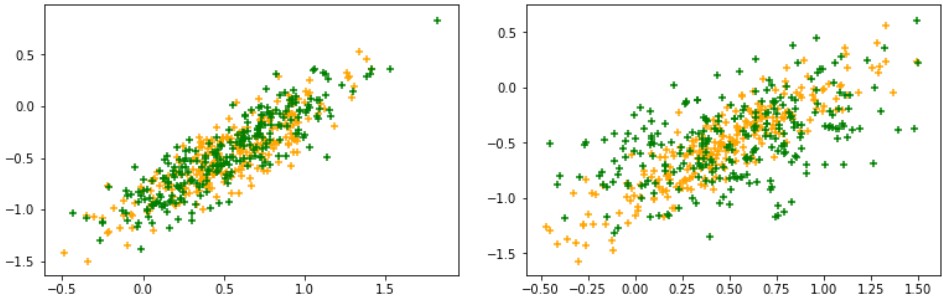

Figure 2: Samples from trained generators. Left: WGAN with quadratic polynomials as discriminator. Right: WGAN with a neural discriminator with 500 neurons.

$\sqrt{2}$, and their standard deviations are all 0.01414. We take $10^5$ samples as training dataset, and 1000 samples as testing dataset. We show one batch (256) of training dataset and the testing dataset in Figure 4. Note that that the density of the ground-truth distribution is highly singular.

We still use Eqn. (33) as the generator for a single Gaussian component. Our generator assume that there are 8 Gaussian components and they have equal weights, and thus our generator does not have any modeling error. The training parameters are eight sets of scaling and biasing parameters in Eqn. (33), each for one Gaussian component.

We first train the model by WGAN with clipping. We use an MLP with 4 hidden layers and relu activations as the discriminator set. We show the result in Figure 5. We can see that the generator's samples are nearly indistinguishable from the real samples. However, the GAN loss and the log likelihood are not consistent. In the initial stage of training, both the **negative** GAN loss and log likelihood are increasing. As the training goes on, the generator's density gets more and more singular, the log likelihood behaves erratically in the latter stage of training. Although the **negative** GAN loss is still increasing, the log likelihood oscillates a lot, and in fact over half of time the log likelihood is $-\infty$. We show the generated samples at intermediate steps in Figure 6, and we indeed see that the likelihood starts to oscillate violently when the generator's distribution gets singular.

This inconsistency between GAN loss and likelihood is observed by other works as well. The reason for this consistency is that the neural discriminators are not a good approximation of the singular density ratios.

We also train the model by maximizing likelihood on the training dataset. We show the result in Figure 7. We can see that the maximal likelihood training got stuck in a local minimum, and failed to exactly recover all 8 components. The log likelihood on training and testing datasets are consistent as expected. Although the log likelihood ($\approx 2.7$) obtained by maximizing likelihood is higher than

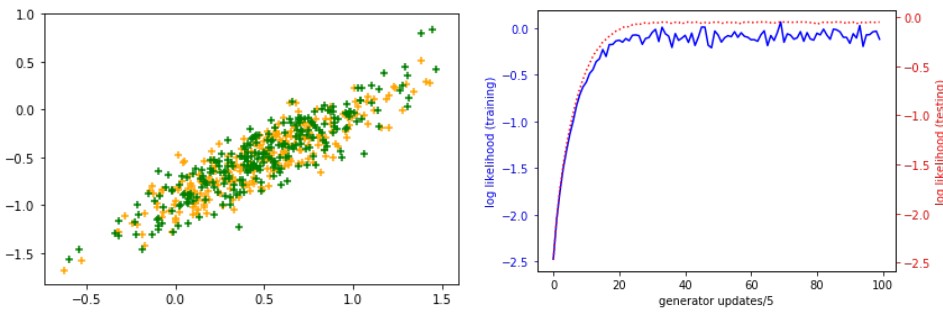

Figure 3: Left: samples from the maximal likelihood estimate. Right: log likelihood on training and testing datasets, trained with SGD.

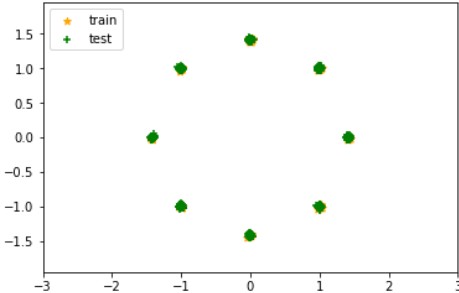

Figure 4: Samples from training and testing datasets.

that ($\approx 2.0$) obtained by WGAN training, its generator is obviously worse than what we obtained in WGAN training. The reason for this is that the negative log-likelihood loss has many local minima, and maximizing likelihood is easy to get trapped in a local minimum.

The FlowGAN (Grover et al., 2017) proposed to combine the WGAN loss and the log likelihood to solve the inconsistency problem. We showed the FlowGAN result on this dataset in Figure 8. We can see that training by FlowGAN indeed makes the training loss and log likelihood consistent. However, FlowGAN got stuck in a local minimum as maximizing likelihood did, which is not desirable.

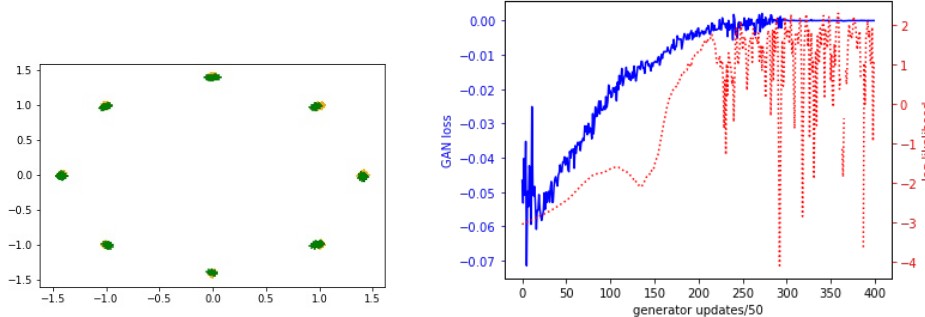

Figure 5: Left: samples from training dataset (yellow) and samples from generator (green). Right: negative GAN loss and log likelihood (evaluated on the testing dataset).

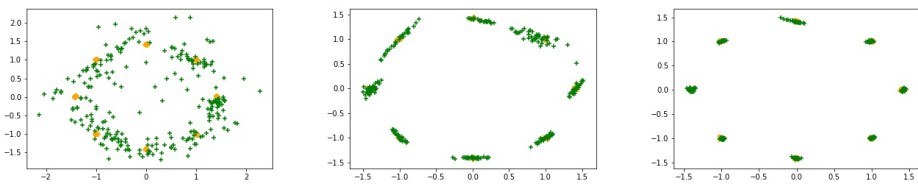

Figure 6: Left to right: generated samples at step 100, 200 and 300, respectively.

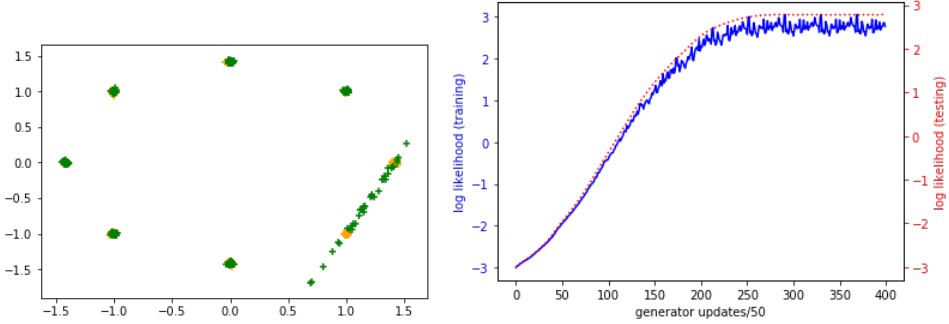

Figure 7: Left: samples from training dataset and samples from generator. Right: log likelihood on training and testing dataset.

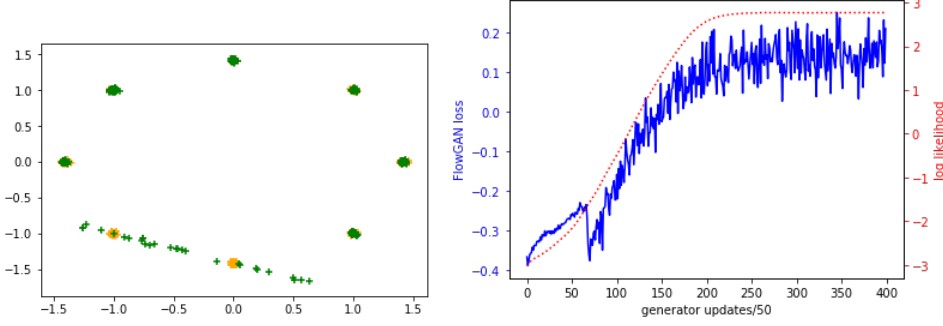

Figure 8: Left: samples from training dataset and samples from generator. Right: negative Flow-GAN loss and log likelihood on testing dataset.

