# OpenReview forum: "On the Discrimination-Generalization Tradeoff in GANs"
_ICLR.cc/2018/Conference — Accept (Poster)_

### Official Review · AnonReviewer1 · 2017-11-23
**Purely theoretical paper on GANs with several novel but not groundbreaking contributions**

**Rating:** 6
**Confidence:** 4

**Review:**

== Paper Summary ==
The paper addresses the problem of balancing capacities of generator and discriminator classes in generative adversarial nets (GANs) from purely theoretical (function analytical and statistical learning) perspective. In my point of view, the main *novel* contributions are:
(a) Conditions on function classes guaranteeing that the induced IPMs are metrics and not pseudo-metrics (Theorem 2.2). Especially I liked an argument explaining why ReLu activations could work better in discriminator that tanh.
(b) Proving that convergence in the neural distance implies a weak convergence (Theorem 2.5)
(c) Listing particular cases when the neural distance upper bounds the so-called bounded Lipschitz distance (also know as the Fortet-Mourier distance) and the symmetrized KL-divergence (Corollary 2.8 and Proposition 2.9).

The paper is well written (although with *many* typos), the topic is clearly motivated and certainly interesting. The related literature is mainly covered well, apart from several important points listed below.

== Major comments ==
In my opinion, the authors are slightly overselling the results. Next I shortly explain why:

(1) First, point (a) above is indeed novel, but not groundbreaking. A very similar result previously appeared in [1, Theorem 5]. The authors may argue that the referenced result deals only with MMDs, that is IPMs specified to the function classes belonging to the Reproducing Kernel Hilbert Spaces. However, the technique used to prove the "sufficient" part of the statement is literally *identical*.

(2) As discussed in the paragraph right after Theorem 2.5, Theorem 10 of [2] presents the same result which is on one hand stronger than Theorem 2.5 of the current paper because it allows for more general divergences than the neural distance and on the other hand weaker because in [2] the authors assumes a compact input space. Overall, Theorem 2.5 of course makes a novel contribution, because the compactness assumption is not required, however conceptually it is not that novel.

(3) In Section 3 the authors discuss the generalization properties of the neural network distance. One of the main messages (emphasized several times throughout the paper) is that surprisingly the capacity of the generator class does not enter the generalization error bound. However, this is not surprising at all as it is a consequence of the way in which the authors define the generalization. In short, the capacity of discriminators (D) naturally enters the picture, because the generalization error accounts for the mismatch between the true data distribution mu (used for testing) and its empirical version hat{mu} (used for training). However, the authors assume the model distribution (nu) is the same both during testing and training. In practice this is not true and during testing GANs use the empirical version of nu. If the authors were to account for this mismatch, capacity of G would certainly pop up as well.

(4) The error bounds of Section 3 are based on a very standard machinery (empirical processes, Rademacher complexity) and to the best of my knowledge do not lead to any new interesting conclusions in terms of GANs.

(5) Finally, I would suggest the authors to remove Section 4. I suggest this mainly because the authors admit in Remark 4.1 that the main result of this section (Theorem 4.1) is a corollary of a stronger result appearing in [2]. Also, the main part of the paper has 13 pages, while a recommended amount is 8.

== Minor comments ==

(1) There are *MANY* typos in the paper. Only few of them are listed below.
(2) First paragraph of page 18, proof of Theorem 2.2. This part is of course well known and the authors may just cite Lemma 9.3.2. of Dudley's "Real analysis and probability" for instance.
(3) Theorem 2.5: "Let ..."
(4) Page 7, "...we may BE interested..."
(5) Corollary 3.2. I doubt that in practice anyone uses discriminator with one hidden unit. The authors may want to consider using the bound on the Rademacher complexity of DNNs recently derived in [3].
(6) Page 8, "..is neural networK"
(7) Page 9: "...interested IN evaluating..."
(8) Page 10. All most ---> almost.

[1] Gretton et al., A Kernel Two-Sample Test, JMLR 2012.
[2] Liu et al, Approximation and Convergence Properties of Generative Adversarial Learning, 2017
[3] Bartlett et al, Spectrally-normalized margin bounds for neural networks, 2017

---

> ### Author Response · Authors · 2018-01-05
> **Thank you for your accurate and insightful comments, part 2**
>
> Reply to your minor comments:
> (1)	We are sorry for this. We carefully corrected typos in the revised version.
> (2)	As you suggested, we now directly cite Lemma 9.3.2. of Dudley's "Real analysis and probability".
> (3)	Thank you for your suggestion. We add the new bound in Appendix A.1 in our revised version. Compared to our previous result for neural network discriminators (Corollary 3.3), the new bound gets rid of the number of parameters, which can be prohibitively large in practice. Moreover, the new bound can be directly applied to the spectral normalized GANs [4], and may explain the empirical success of the spectral normalization technique.
>
> [1] Liu et al, Approximation and Convergence Properties of Generative Adversarial Learning, 2017
> [2] Ivo Danihelka, Balaji Lakshminarayanan, Benigno Uria, DaanWierstra, and Peter Dayan. Comparison of maximum likelihood and gan-based training of real nvps. arXiv preprint arXiv:1705.05263, 2017.
> [3] Aditya Grover, Manik Dhar, and Stefano Ermon. Flow-gan: Bridging implicit and prescribed learning in generative models. arXiv preprint arXiv:1705.08868, 2017.
> [4] Anonymous. Spectral normalization for generative adversarial networks. International Conference on Learning Representations, 2018. URL https://openreview.net/forum?id=B1QRgziT-.

---

> ### Author Response · Authors · 2018-01-05
> **Thank you for your accurate and insightful comments, part 1**
>
> Thank you very much for your insightful review. Your comments help us improve the paper a lot! Overall, all your comments are correct. The following are our replies to some details of your comments.
>
> Major comments:
> (1)	Yes, for the sufficient part, the proof is standard and is the same as that of the uniqueness of weak convergence. The essential idea is in fact to make use of the moment matching effect of GANs, which is “obvious” for neural distance but tricky for neural divergence. As long as we have moment matching, we now directly cite Lemma 9.3.2. of Dudley's "Real analysis and probability" as you suggested.
>
> (2)	Theorem 2.5 and its proof have two important differences from Theorem 10 of [1]. First, as you pointed out, it gets rid of the compactness assumption. Second, its proof (in Appendix E) in fact gives the convergence rate of the GAN training as the neural distance is minimized. We can see that the convergence rate depends both on the optimization error decay rate ( d_F(\mu, \nu_n) ) and the representation error decay rate (\epsilon(r) defined in Proposition 2.7). This convergence rate provides guidance to improve the training speed of GANs, by utilizing either faster optimization algorithms or more representative discriminator set. On the contrary, the existence proof in [2] does not provide an estimate of the convergence rate.
>
> (3)	Yes, you are correct.  In the revised version, we emphasize that only when evaluated with neural distance, generalization is guaranteed as long as the discriminator set is small enough, regardless of the size of the generator or hypothesis set. In this paper, we want to bound $d_F(\mu, nu_m)$, i.e., the difference between the unknown target distribution \mu and the learned distribution \nu_m, instead of the testing error $d_F(\hat{\mu}, \hat{nu}_m)$. If the the testing error, the capacity of the generator set would certainly pop up, as you commented.
>
> (4)	The error bound under neural distance in Section 3 (Theorem 3.1) is indeed based on a very standard machinery (empirical processes, Rademacher complexity). However, we think itself and its induced results are still valuable for two reasons.
>
> First, although its derivation is standard, its meaning is very different from that in supervised learning. When the evaluation metric is taken as neural distance, the generalization error can be purely bounded by the complexity of the discriminator set. This seemingly loose bound is indeed tight *in terms of the order of sample size m* for several common cases. For example, for GANs with neural discriminators and for MMD-GANs, the bound is O(m^{-1/2}); for Wasserstein distance, the bound is O(m^{-1/d}); for total variation distance, the bound is O(1). In all these cases, the bound is indeed tight *in terms of the order of sample size m*. Of course, the bound is still very loose in other aspects, due to the ignorance of the generator set.
>
> Second, we use our bounds between neural distance and other standard measures to derive generalization for other evaluation metrics. Especially, when the KL divergence is used as the evaluation metric, our bound (Corollary 3.5) suggests that the generator and discriminator sets must be compatible in that the log density ratios of the generators and the true distributions should exist and be included inside the linear span of the discriminator set. The strong condition that log-density ratio should exist partially explains the counter-intuitive behavior of testing likelihood in flow GANs ([2,3]).
>
> 5.	We move the neural divergence section to Appendix B, only summarizing our new contributions in discrimination properties of f-GANs in Remark 2.2. We would like to point out that the following result is our novel contribution: a neural $f$-divergence is discriminative if linear span of its discriminators \emph{without the output activation function} is dense in the bounded continuous function space. Both its statement and its proof are nontrivial and cannot be found in other places.

---

### Official Review · AnonReviewer3 · 2017-11-27
**This paper provides a mathematical analysis of the role of the size of the adversary/discriminator set in GANs.  It argues that on the one hand, large discriminator sets are useful as they help isolate the target distribution; on the other hand, small discriminator sets help with small sample effects.  Unfortunately, I seem to have found some flaws.**

**Rating:** 3
**Confidence:** 4

**Review:**

In more detail, the analysis of the paper is as follows.  Firstly, it primarily focuses on GAN objective functions which are "integral probability metrics (IPMs)"; one way to define these is by way of similarity to the W-GAN, namely IPMs replace the 1-Lipschitz functions in W-GAN with a generic set of functions F.  The paper overall avoids computational issues and treats the suprema as though exactly solved by sgd or related heuristic (the results of the paper simply state supremum, but some of the prose seems to touch on this issue).

The key arguments of the paper are as follows.

1. It argues that the discriminator set should be not simply large, it should be dense in all bounded continuous functions; as a consequence of this, the IPM is 0 iff the distributions are equal (in the weak sense).  Due to this assertion, it says that it suffices to use two layer neural networks as the discriminator set (as a consequence of the "universal approximation" results well-known in the neural network literature).

2. It argues the discriminator set should be small in order to mitigate small-sample effects.  (Together, points 1 and 2 mimic a standard bias-variance tradeoff in statistics.)  For this step, the paper relies upon standard Rademacher results plus a little bit of algebraic glue.  Curiously, the paper chooses to argue (and forms as a key tenet, indeed in the abstract) that the size of the generator set is irrelevant for this, only the size of the discriminator matters.

Unfortunately, I find significant problems with the paper, in order from most severe to least severe.

A.  The calculation ruling out the impact of the generator in generalization calculations in 2 above is flawed.  Before pointing out the concrete bug, I note that this assertion runs completely counter to intuition, and thus should be made with more explanation (as opposed to the fortunate magic it is presented as).  Moreover, I'll say that if the authors choose to "fix" this bug by adding a generator generalization term, the bound is still a remedial application of Rademacher complexity, so I'm not exactly blown away.  Anyway, the bug is as follows.  The equation which drops the role of the generator in the generalization calculation is the equation (10).  The proof of this inequality is at the start of appendix E.  Looking at the derivation in that appendix, everything is correct up to the second-to-last display, the one with a supremum over nu in G.  First of all, this right hand side should set off alarm bells; e.g., if we make the generator class big, we can make this right hand side essentially as big as the IPM allows even when mu = mu_m.  Now the bug itself appears when going to the next display: if the definition of d_F is expanded, one obtains two suprema, each own over _their own_ optimization variable (in this case the variables are discriminator functions).  When going to the next equation, the authors accidentally made the two suprema have the same variable and invoke a fortuitous but incorrect cancellation.  As stated a few sentences back, one can construct trivial counterexamples to these inequalities, for instance by making mu and mu_m arbitrarily close (even exactly equal if you wish) and then making nu arbitrarily far away and the discriminator set large enough to identify this.

B. The assertions in 1, regarding sizes of discriminator sets needed to achieve the goal of the IPM being 0 iff the distributions are equal (in the weak sense), are nothing more than immediate corollaries of approximation results well-known for decades in the neural network literature.  It is thus hard to consider this a serious contribution.

C. I will add on a non-technical note that the paper's assertion on what a good IPM "should be" is arguably misled.  There is not only a meaning to specific function classes (as with Lip_1 in Wasserstein_1) beyond simply "many functions", but moreover there is an interplay between the size of the generator set and the size of the discriminator set.  If the generator set is simple, then the discriminator set can also get away with being simple (this is dicussed in the Arora et al 2017 ICML paper, amongst other places).  Perhaps I am the one that is misled, but even so the paper does not appear to give a good justification of its standpoint.

I will conclude with typos and minor remarks.  I found the paper to contain a vast number of small errors, to the point that I doubted a single proofread.

Abstract, first line: "a minimizing"?  general grammar issue in this sentence; this sort of issue throughout the paper.

Abstract, "this is a mild condition".  Optimizing over a function class which is dense in all bounded measurable functions is not a mild assumption.  In the particular case under discussion, the size of the network can not be bounded (even though it has just two layers, or as the authors say is the span of single neurons).

Abstract, "...regardless of the size of the generator or hypothesis set".  This really needs explanation in the abstract, it is such a bold claim.  For instance, I wrote "no" in the margin while reading the abstract the first time.

Intro, first line: its -> their.

Intro, #3 "energy-based GANs": 'm' clashes with sample size.

Intro, bottom of page 1, the sentence with "irrelenvant": I can't make any sense of this sentence.

Intro, bottom of page 1, "is a much smaller discriminator set": no, the Lip_1 functions are in general incomparable to arbitrary sets of neural nets.

From here on I'll comment less on typos.

Middle of page 2, point (i): this is the only place it is argued/asserted that the discriminator set should contain essentially everything?  I think this needs a much more serious justification.

Section 1.1: Lebegure -> Lebesgue.

Page 4, vicinity of equation 5: there should really be a mention that none of these universal approximation results give a meaningful bound on the size of the network (the bound given by Barron's work, while nice, is still massive).

Start of section 3.  To be clear, while one can argue that the Lipschitz-1 constraint has a regularization effect, the reason it was originally imposed is to match the Kantorovich duality for Wasserstein_1.  Moreover I'll say this is another instance of the paper treating the discriminator set as irrelevant other than how close it is to being dense in all bounded measurable functions.

---

> ### Author Response · Authors · 2017-12-06
> **Reply to your other minor remarks**
>
> Abstract, "...regardless of the size of the generator or hypothesis set". This really needs explanation in the abstract, it is such a bold claim. For instance, I wrote "no" in the margin while reading the abstract the first time.
>
> Our reply: Hope that you are convinced by this *bold* claim now.
>
> Page 4, vicinity of equation 5: there should really be a mention that none of these universal approximation results give a meaningful bound on the size of the network (the bound given by Barron's work, while nice, is still massive).
>
> Our reply: In the revised version, we will mention that most early universal approximation results, e.g., Cybenko, 1989; Hornik et al., 1989; Hornik, 1991; Leshno et al., 1993, are qualitive results and do not give a meaningful approximation rates. Barron (1993) and Bach (2017) give the approximation rates of two-layer neural networks.
>
> Start of section 3. To be clear, while one can argue that the Lipschitz-1 constraint has a regularization effect, the reason it was originally imposed is to match the Kantorovich duality for Wasserstein_1. Moreover, I'll say this is another instance of the paper treating the discriminator set as irrelevant other than how close it is to being dense in all bounded measurable functions.
>
> Our reply: We agree with you on the initial motivation of the Lipschitz-1 constraint in WGAN. We do not require that the discriminator set be dense in C_b(X).
> On one hand, our basic requirement is that span of the discriminator set is dense in C_b(X). The larger the discriminator set is, the more discriminative the IPM will be. On the other hand, the smaller the discriminator set is, the smaller the generalization error will be. As our title indicates, there is a discrimination-generalization tradeoff in GANs. Fortunately, we show that several GANs *in practice* already chose their discriminator set at the sweet point.

---

> ### Author Response · Authors · 2017-12-06
> **You may have misundertood our results on discrimination; reply to your minor remarks**
>
> Abstract, "this is a mild condition". Optimizing over a function class which is dense in all bounded measurable functions is not a mild assumption. In the particular case under discussion, the size of the network cannot be bounded (even though it has just two layers, or as the authors say is the span of single neurons).
>
> Our reply: This is a *big misunderstanding* of our results. Our Theorem 2.2 says that it suffices to optimizing over a function class F, whose span is dense in all bounded measurable functions, to guarantee the discriminative power of GANs. The optimization is over *F*, not *span of F*.
> We emphasize the *span* several times in our paper. What’s the difference the *span* makes?
> 1.	For tanh (or sigmoid) activation: neural networks with one hidden layer and *sufficiently many neurons* can approximate any continuous functions; neural networks with one hidden layer and *only one neuron* are sufficient to discriminate any two distributions.
> 2.	For relu activation: neural networks with one hidden layer, *sufficiently many neurons* and *unbounded weights* can approximate any continuous functions; neural networks with one hidden layer, *only one neuron* and *bounded weights* are sufficient to discriminate any two distributions.
> 3.	A simpler example: the span of two points (1,0) and (0,1) is dense in the whole R^2 plane!
> Therefore, our condition is indeed a very mild condition.
>
> Intro, bottom of page 1, the sentence with "irrelenvant": I can't make any sense of this sentence.
>
> Our reply: As we listed in the first page, different GANs define their objective functions (e.g., Wasserstein distance and f-divergence) with different *non-parametric* discriminator sets, while in practice they use parametric discriminator sets as surrogates, which leads to objective functions like neural distance and neural divergence. In this sentence, we say that the properties of their objective functions in mind and objective functions in practice can be fundamentally different or even irrelevant.  The main goal of this paper is to close this gap, i.e., to provide discrimination and generalization properties for practical GANs, which use parametric discriminator sets.
>
> Intro, bottom of page 1, "is a much smaller discriminator set": no, the Lip_1 functions are in general incomparable to arbitrary sets of neural nets.
>
> Our reply: WGAN is motivated to optimized over all Lip_1 functions. In practice, WGAN optimizes over neural networks with bounded parameters (weight clipping). It is argued in the WGAN paper that this practical discriminator set is contained in Lip_K function class for certain K>0. However, the set of neural networks with bounded parameters is a much smaller set compared to the Lip_K function class.
> We note that different GAN variants will also use different parametric function classes. We use F_{nn} as an abstract symbol for all these parametric discriminator sets.
>
> Middle of page 2, point (i): this is the only place it is argued/asserted that the discriminator set should contain essentially everything? I think this needs a much more serious justification.
>
> Our reply: We did not argue/assert that that the discriminator set should contain essentially everything. We argue that the span of the discriminator set should be dense in bounded continuous function space. As we pointed out before, the *span* makes a big difference.

---

> ### Author Response · Authors · 2017-12-06
> **The flaws you pointed out in fact are not flaws; Reply to your "significant problems"**
>
> Thank you for your review! We can see that you went to some details of the paper, and we are grateful for that. However, you may have some misunderstandings of the main results as we will elaborate as below.
>
> A.	In fact, our derivations and results on generalization is correct. The flaw you found in Appendix E (Proof of Equation 10) is this derivation:
>  |d_F(u, v) – d_F(u_m, v)| <= d_F(u, u_m) := \sup_{f\in F} E_{u}[f] – E_{u_m}[f].
> In fact, this is purely the triangle inequality of the pseudo metric d_F. It can also be proved from the definition of d_F as follows:
> d_F(u, v) – d_F(u_m, v)
> := \sup_{f\in F} ( E_{u}[f] – E_{v}[f] ) - \sup_{g\in F} ( E_{u_m}[g] – E_{v}[g] )
> \le \sup_{f\in F} ( E_{u}[f] – E_{v}[f] - E_{u_m}[f] + E_{v}[f] )
> = \sup_{f\in F} ( E_{u}[f] - E_{u_m}[f] ) =: d_F(u, u_m)
> Symmetrically, we can prove d_F(u_m, v) – d_F(u, v) \le D_F(u_m, u) . Therefore, we have proved |d_F(u, v) – d_F(u_m, v)| <= d_F(u, u_m).
> We can make the two variables (i.e., f and g) into one variable (i.e., f) because $- \sup_{g\in F} ( E_{u_m}[g] – E_{v}[g] ) \le -  E_{u_m}[f] – E_{v}[f]$ for any $f \in F$.
>
> B.	Our main contributions in discriminative power of GANs are:
> (1)	We give the necessary and sufficient condition for discrimination: span(F) is dense in bounded continuous function space; see Theorem 2.2.
> (2)	For any metric space, minimizing GAN’s objective function implies weak convergence; see Theorem 2.6.
> These two theorems have nothing to do with the universal approximation property of neural networks. Therefore, it is unfair to say that they are *immediate corollaries of approximation results well-known for decades in the neural network literature*.
> Combining our Theorem 2.2 and the well-known universal approximation property of neural networks, we proved that GANs with neural works as discriminators are discriminative; see Theorem 2.3 and Corollary 2.4.
>
> C.	From the perspective of *game theory*, there is an interplay between the generator set and the discriminator set for the existence of equilibria; see Arora (2017). From the perspective of *minimizing a loss function* (e.g., neural distance/divergence), the existence of a global minimum is a straightforward result from continuity of the loss function and compactness of the hypothesis set. In this paper, we analyze the properties of different loss functions, regardless of the hypothesis set. The analog in supervised learning is analyzing properties of different loss functions (negative log-likelihood, mean square error, regularized or not), regardless of the hypothesis set. From this perspective of *minimizing a loss function*, the goodness of the discriminator set (defines the loss function) can be studied independently from the generator set.
> In this paper, except our results on KL divergence (i.e., Proposition 2.9 and Corollary 3.5), we do not make any assumptions on the generator set. This makes our results widely applicable, regardless of the user’s choice of the generator set. Of course, given the loss function, one can study the accuracy and generalization for a specific generator set, as we commented several times in our paper.

---

### Official Review · AnonReviewer4 · 2017-12-14
**An interesting insight to the GANs, especially the discriminators**

**Rating:** 7
**Confidence:** 4

**Review:**

The authors provide an insight into the discriminative and generalizable aspect of the discriminator in GANs. They show that the richer discriminator set to enhance the discrimination power of the set while reducing the generalization bound. These facts are intuitive, but they made a careful analysis of it.

The authors provide more realistic analysis of discriminators by relaxing the constraint on discriminator set to have a richer closure of linear span instead of being rich by itself which is suitable for neural networks.

They analyze the weak convergence of probability measure under neural distance and generalize it to the other distances by bounding the neural distance.

For the generalization, they follow the standard generalization procedure and techniques while carefully adapt it to their setting.

Generally, I like the way the authors advertise their results, but it might be a bit oversold, especially for readers with theory background.

The authors made a good job in clarifying what is new and what is borrowed from previous work which makes this paper more interesting and easy to read.

Since the current work is a theoretical work, being over 8 pages is acceptable, but since the section 4 is mostly based on the previous contributions, the authors might consider to have it in the appendix.

---

> ### Author Response · Authors · 2018-01-05
> **Replies to your comments and changes in the revised version**
>
> We thank for your insightful comments and your appreciation of our results.
>
> Thank you for pointing out that we might oversell the results, especially for readers with theory background. We avoid this in the revised version in two ways. First, we simplify the proof of standard results (the proof of the sufficient part of Theorem 2.1 and the proof of Theorem 3.1) and focus more on their implications on GANs. Second, to avoid misunderstanding, we emphasize that only when evaluated with neural distance, generalization is guaranteed as long as the discriminator set is small enough, regardless of the size of the generator or hypothesis set. We explain that this seemingly-surprising result is reasonable because the evaluate metric (neural distance) is defined by the discriminator set and is “weak” compared to standard metrics like BL distance and KL divergence.
>
> We move the neural divergence section to Appendix B, only summarizing our new contributions in discrimination properties of f-GANs in Remark 2.2. We would like to point out that the following result is our novel contribution: a neural $f$-divergence is discriminative if linear span of its discriminators \emph{without the output activation function} is dense in the bounded continuous function space. Both its statement and its proof are nontrivial and cannot be found in other places.
>
> Finally, in the revised version, we add one generalization bound for GANs with DNNs as discriminators in Appendix A.1. This bound makes use of the recent result on Rademacher complexity of DNNs in Bartlett et al. (2017). Compared to our previous result for neural network discriminators (Corollary 3.3), the new bound gets rid of the number of parameters, which can be prohibitively large in practice. Moreover, the new bound can be directly applied to the spectral normalized GANs (Anonymous, 2018), and may explain the empirical success of the spectral normalization technique.
>
> [Bartlett et al., 2017)] Peter L Bartlett, Dylan J Foster, and Matus J Telgarsky. Spectrally-normalized margin bounds for neural networks. In Advances in Neural Information Processing Systems, pp. 6241–6250, 2017.
> [Anonymous, 2018] Anonymous. Spectral normalization for generative adversarial networks. International Conference on Learning Representations, 2018. URL https://openreview.net/forum?id=B1QRgziT-.

---

### Public Comment · (anonymous) · 2017-11-28
**Some questions**

I'm going over papers on theoretical results on GANs and got into this one.

In the first part of the paper, the author asks whether the neural distance is "discriminative" or not, that is, whether being equal in neural distance implies that the two distributions are actually identical. It is shown that the answer is affirmative for a large class of discriminators, including neural networks. Based on this property, it is shown that the learned distribution weakly converge to the target distribution. I found that the proof for this "discriminative" result seems like a straightforward exercise in real analysis. I wonder whether this result has been discovered before.

The generalization part seems reasonable, and shares several similarities with supervised learning. A drawback of this paper is that they do not consider the impact of training on GANs, which matters a lot in practice.

---

> ### Author Response · Authors · 2017-11-29
> **the uniqueness of our discrimination results**
>
> Thank you for your interest, questions and comments!
>
> First, as far as we know, no previous work gives a sufficient and necessary conditions of the discriminator set under which the neural distance is discriminative. In previous work, the discriminative power of GANs is typically justified by assuming that the discriminator set has enough capacity, such as all functions taking values in [0,1] in vanilla GAN or all Lipchitz functions with Lipchitz constant 1 in WGAN. However, we use neural networks with bounded parameters in practice. The main goal of our results is to close this gap between previous theoretical results and practices.  We'd like to mention the following points on the discriminative power.
> (1). [1] and [2] also noticed that GANs with restricted discriminators may not be discriminative, but this problem was addressed in neither [1] nor [2].
> (2). Previous work also uses the universal approximation property of neural networks to justify the discriminative power empirically. Our results show that the neural distance is discriminative under much weaker condition, that is, span of the discriminator set can approximate any continuous functions. This justifies why neural network with bounded parameters works in practice.
> (3). Our discriminative results also apply to neural divergence (Theorem 4.1), which requires that span of the discriminators without the nonlinear activation in the last layer is dense in bounded continuous functions. This coincides with the implementation difference between vanilla GAN and WGAN, where WGAN simply uses discriminators in the vanilla GAN without the nonlinear activation in the last layer.
>
> Yes, the generalization part shares similarities with the supervised learning. However, the most important difference is that: in supervise learning, the complexity of the hypothesis set (G) bounds the generalization error; in GANs, the complexity of the discriminator set (F) bounds the generalization error, which can be independent of the hypothesis set.
>
> We agree that we do not consider the impact of training on GANs, which is very important in practice. We noticed several recent papers are working in this direction, e.g., [3, 4]. We are currently working on stabilizing the training of GANs through our approach.
>
> [1] Sanjeev Arora, Rong Ge, Yingyu Liang, Tengyu Ma, and Yi Zhang. Generalization and equilibrium in generative adversarial nets (gans). arXiv preprint arXiv:1703.00573, 2017.
> [2] Shuang Liu, Olivier Bousquet, and Kamalika Chaudhuri. Approximation and convergence properties
> of generative adversarial learning. arXiv preprint arXiv:1705.08991, 2017.
> [3] Jerry Li, Aleksander Madry, John Peebles, and Ludwig Schmidt. Towards understanding the dynamics of generative adversarial networks. arXiv preprint arXiv:1706.09884, 2017b.
> [4] Martin Heusel, Hubert Ramsauer, Thomas Unterthiner, Bernhard Nessler, G¨unter Klambauer, and Sepp Hochreiter. Gans trained by a two time-scale update rule converge to a nash equilibrium. arXiv preprint arXiv:1706.08500, 2017.

---

### Author Response · Authors · 2018-01-05
**Changes made to the paper**

We thank all the reviewers for their insightful comments. Based on their feedbacks, we made the following changes to the paper.

1.We move the neural divergence section to Appendix B, only summarizing our new contributions in discrimination properties of f-GANs in Remark 2.2.

2.We add one generalization bound for GANs with DNNs as discriminators in Appendix A.1. This bound makes use of the recent result on Rademacher complexity of DNNs in Bartlett et al. (2017). Compared to our previous result for neural network discriminators (Corollary 3.3), the new bound gets rid of the number of parameters, which can be prohibitively large in practice. Moreover, the new bound can be directly applied to the spectral normalized GANs (Anonymous, 2018), and may explain the empirical success of the spectral normalization technique.

3.We emphasize that only when evaluated with neural distance, generalization is guaranteed as long as the discriminator set is small enough, regardless of the size of the generator or hypothesis set. We explain that this seemingly-surprising result is reasonable because the evaluate metric (neural distance) is defined by the discriminator set and is “weak” compared to standard metrics like BL distance and KL divergence.

4.We make other small changes according to reviews, and correct typos in the original draft.

[Bartlett et al., 2017)] Peter L Bartlett, Dylan J Foster, and Matus J Telgarsky. Spectrally-normalized margin bounds for neural networks. In Advances in Neural Information Processing Systems, pp. 6241–6250, 2017.
[Anonymous, 2018] Anonymous. Spectral normalization for generative adversarial networks. International Conference on Learning Representations, 2018. URL https://openreview.net/forum?id=B1QRgziT-.

---

### Decision · Program_Chairs · 2018-01-29
**ICLR 2018 Conference Acceptance Decision**

**Decision:**

Accept (Poster)

**Comment:**

I recommend acceptance. The two positive reviews point out the theoretical contributions. The authors have responded extensively to the negative review and I see no serious flaw as claimed by the negative review.